# STRA8–RB interaction is required for timely entry of meiosis in mouse female germ cells

Ryuki Shimada [1], Yuzuru Kato[2], Naoki Takeda[3], Sayoko Fujimura[4], Kei-ichiro Yasunaga[4], Shingo Usuki[4], Hitoshi Niwa[5], Kimi Araki [3,6] & Kei-ichiro Ishiguro [1] ✉

Meiosis is differently regulated in males and females. In females, germ cells initiate meiosis within a limited time period in the fetal ovary and undergo a prolonged meiotic arrest until puberty. However, how meiosis initiation is coordinated with the cell cycle to coincide with S phase remains elusive. Here, we demonstrate that STRA8 binds to RB via the LXCXE motif. Mutation of the RB-binding site of STRA8 in female mice delays meiotic entry, which consequently delays progression of meiotic prophase and leads to precocious depletion of the oocyte pool. Single-cell RNA-sequencing analysis reveals that the STRA8–RB interaction is required for S phase entry and meiotic gene activation, ensuring precise timing of meiosis initiation in oocytes. Strikingly, the results suggest STRA8 could sequester RB from E2F during pre-meiotic G1/S transition. This study highlights the gene regulatory mechanisms underlying the female-specific mode of meiotic initiation in mice.

Meiotic entry coincides with the pre-meiotic S phase and is followed by meiotic prophase, a prolonged G2 phase involving numerous meiosis-specific chromosome events[1–3]. In mouse germ cells, meiotic entry is triggered by the co-expression of stimulated by retinoic acid 8 (STRA8)[4–8] and meiosis initiator (MEIOSIN)[9] in the embryonic ovary and postnatal testis. MEIOSIN, together with STRA8, binds to and activates meiotic genes that are essential for meiosis-specific events and plays a crucial role in meiotic initiation in both male and female germ cells[9,10].

Interestingly, the timing of meiosis onset is different in males and females, coordinated with the developmental process of germ cells. In fetal testes, male germ cells stop proliferating and enter G0/G1 arrest at E14.5[11–13]. Meiotic entry is accompanied by the differentiation of B-type spermatogonia into spermatocytes at puberty and continuously occurs in the testis throughout life. In fetal ovaries, female germ cells quit mitotic proliferation and enter meiosis at E13.5–E14.5. In fetal ovaries, meiotic initiation is restricted to a specific period

around E14.5, concomitantly with sex determination[14–16] and down-regulation of pluripotency genes, such as *Oct4*, *Sox2*, and *Nanog*[17–20]. In fetal ovaries, meiosis is initiated with radial wave geometry, which propagates along the anterior to posterior gonadal pole, and inter-cellular bridges of oocytes coordinate the transition from pluripotency to meiosis via the dilution of regulatory factors[21]. Retinoic acid (RA) signaling is one of the key pathways involved in meiotic initiation in the fetal ovary, ensuring the timely expression of *Stra8* and meiotic genes[15,14,22]. The persistence of meiosis despite delayed expression of STRA8 in fetal ovaries lacking aldehyde dehydrogenase 1 A (ALDH1A) proteins (ALDH1A1, 2 and 3)[23–25] or RARs[26] suggests the involvement of other signaling mechanisms in meiotic initiation. In addition to RA signaling, in vitro study suggested that meiotic initiation is also regulated by several regulatory signals in female PGC-like cell (PGCLC). Bone morphogenetic protein (BMP), synergistically with RA, primes the competency of female germ cells for meiotic initiation[27], and ZGLP1 activates the key genes of oogenic processes, including meiosis[28].

[1]Department of Chromosome Biology, Institute of Molecular Embryology and Genetics (IMEG), Kumamoto university, Honjo 2-2-1, Chuo-ku, Kumamoto, Kumamoto 860-0811, Japan. [2]Mammalian Development Laboratory, Department of Gene Function and Phenomics, National Institute of Genetics, Mishima, Shizuoka 411-8540, Japan. [3]Institute of Resource Development and Analysis, Kumamoto University, Kumamoto 860-0811, Japan. [4]Liaison Laboratory Research Promotion Center, IMEG, Kumamoto University, Kumamoto 860-0811, Japan. [5]Department of Pluripotent Stem Cell Biology, IMEG, Kumamoto university, Honjo 2-2-1, Chuo-ku, Kumamoto, Kumamoto 860-0811, Japan. [6]Center for Metabolic Regulation of Healthy Aging, Kumamoto University, 1-1-1, Honjo, Kumamoto 860-8556, Japan. ✉e-mail: ishiguro@kumamoto-u.ac.jp

Female germ cells fail to enter the pre-meiotic S phase in *Stra8* knockout (KO) mice[4]. However, the mechanism by which the initiation of meiosis coordinates with the cell cycle to coincide with the pre-meiotic S phase, which has a substantially longer duration than the mitotic S phase in other cell types, remains unknown[29]. Furthermore, in fetal ovaries, primary oocytes undergo long-term G2 arrest in the middle of the meiotic prophase that persists until puberty. Therefore, the reproductive lifespan of females is conferred by the oocyte pool that initiates meiosis during this limited time window in fetal ovaries. The precise mechanisms by which meiotic initiation and subsequent prophase arrest are regulated during the development of fetal ovaries remain elusive.

In this study, we investigated the female-specific mechanisms of meiotic initiation in mice. We found that STRA8 binds to Retino-blastoma (RB) family proteins (RB1 and p107) independent of MEIO-SIN. Genetic and single-cell RNA-sequencing (scRNA-seq) analyses demonstrate that the STRA8–RB interaction in female germ cells is crucial for timely progression into the pre-meiotic S phase and timely entry into meiosis. Our data suggest that STRA8 sequesters RB from E2F to promote pre-meiotic G1/S transition in female germ cells. Our findings highlight the meiotic cell cycle mechanism in female germ cells.

## Results

### STRA8 interacts with the RB family proteins

Previously, we identified MEIOSIN as a STRA8-interacting protein by immunoprecipitation (IP) of chromatin extracts of mouse testes, followed by mass spectrometry (MS) analysis[9]. Intriguingly, in addition to MEIOSIN, retinoblastoma family proteins, RB1 and p107/RB-Like1 (hereafter referred to as RB for RB1 and p107, unless otherwise stated), were also identified as STRA8-interacting proteins[9]. In the canonical cell cycle, RB family proteins bind to and suppress E2F transcription factors that play a pivotal role in S phase progression by activating multiple cell cycle-related genes[30,31]. p107 plays a more specific role by binding to E2F4 or E2F5 to form the transcriptional repressor DREAM complex together with MuvB core proteins (Lin9, Lin37, Lin52, Lin54) and represses both G1/S and G2/M cell cycle-related genes[32,33]. However, the mechanism by which RB family proteins participate in the meiotic cell cycle in germ cells remains elusive.

Notably, the C-terminus of STRA8 possesses the LXCXE motif that is known to bind to RB families[34], and the potential LXCXE motif is conserved in vertebrate homologs of STRA8 (Supplementary Fig. 1). To address the physiological meaning of the interaction between STRA8 and RB family proteins (RB1 and p107), we generated *Stra8* [ΔLXCXE]-*3xFLAG-HA-p2A-GFP* knock-in (*Stra8* [ΔLXCXE]-*3FH* KI) mice, in which alanine substitutions were introduced at the Leu, Cys, and Glu residues in the LXCXE motif of STRA8, so that LKCLE residues were changed to AKALA residues in the mutant (Fig. 1a). Tandem affinity purification with anti-FLAG and anti-HA antibodies followed by western blotting using testis chromatin extracts from *Stra8* [wt]-*3xFLAG-HA-p2A-GFP* knock-in (*Stra8* [wt]-*3FH* KI) mice demonstrated that FLAG-HA-tagged wild-type STRA8 co-immunoprecipitated with RB1 and p107 as well as MEIOSIN (Fig. 1b). In contrast, the FLAG-HA-tagged mutant STRA8 lacking the LXCXE motif failed to co-immunoprecipitate with RB1 and p107, while the STRA8-MEIOSIN interaction was intact (Fig. 1b). These results suggest that STRA8 interacts with RB1 and p107 via the LXCXE motif. To examine whether RB1, STRA8, and MEIOSIN formed a tertiary complex, the anti-FLAG IP product from *Stra8* [wt]-*3FH* KI testis was subjected to second IP using the anti-RB1 antibody. Intriguingly, RB1 co-immunoprecipitated with STRA8, but not with MEIOSIN (Fig. 1c), suggesting that STRA8 binds to RB1 independently of MEIOSIN. This suggests that STRA8 forms two distinct subcomplexes: one with MEIOSIN and the other with RB1. Although we do not know the exact reason, one possibility is that binding of RB1 or MEIOSIN may exclude

interaction with another due to steric hindrance, which awaits further structural analysis of the STRA8 complexes.

Note that although we attempted to immune-precipitate STRA8 from fetal ovaries, it was technically difficult to show the STRA8–RB1 interaction biochemically by IP-WB in fetal ovaries because of the limited starting materials of fetal ovaries.

### Nuclear localization of STRA8 depends on RB in spermatogonia and oogonia

During spermatogenesis, the first expression of *Stra8* occurs in differentiating spermatogonia, and the second expression occurs in pre-leptotene spermatocytes in response to RA[6,35–37]. In pre-leptotene spermatocytes, MEIOSIN plays a role in the retention of STRA8 in the nuclei[9]. In differentiating spermatogonia, where MEIOSIN is not expressed, STRA8 localization was skewed to the cytoplasm rather than the nuclei in *Stra8* [ΔLXCXE]-*3FH* KI homozygous (*Stra8* [ΔLXCXE/ΔLXCXE]) mice, whereas it was largely localized to the nuclei in wild-type and *Meiosin* KO mice (Supplementary Fig. 2a), suggesting that RB is required for the retention of STRA8 in the nuclei of differentiating spermatogonia. In contrast, in pre-leptotene spermatocytes, STRA8 was largely localized to the nuclei in *Stra8* [ΔLXCXE/ΔLXCXE] as well as in wild type, but not in *Meiosin* KO mice (Supplementary Fig 2b), suggesting that RB, but not MEIOSIN, is dispensable for the nuclear retention of STRA8 in pre-leptotene spermatocytes. Thus, the nuclear localization of STRA8 depends on two independent pathways: RB-mediated nuclear localization of STRA8 in differentiating spermatogonia and MEIOSIN-mediated nuclear localization of STRA8 in pre-leptotene spermatocytes. Curiously, however, although STRA8–RB interaction was observed in the testes, *Stra8* [ΔLXCXE/ΔLXCXE] males exhibited seemingly normal fertility with no apparent defects in adult testes and epididymis (Supplementary Fig. 2c–e), suggesting that the STRA8–RB interaction was dispensable for spermatogenesis.

In embryonic ovaries at E14.5, FLAG-HA-tagged STRA8 was expressed from both *Stra8* [wt]-*3FH* KI and *Stra8* [ΔLXCXE]-*3FH* KI alleles (Fig. 1d). Importantly, however, the localization of FLAG-HA-tagged STRA8[ΔLXCXE] protein produced from *Stra8* [ΔLXCXE]-*3FH* KI allele was skewed in the cytoplasm compared to the nuclei, whereas FLAG-HA-tagged wild-type STRA8 produced from *Stra8* [wt]-*3FH* KI allele localized to the nuclei as well as the cytoplasm (Fig. 1d). Although MEIOSIN was expressed in these heterozygous *Stra8* [wt]-*3FH* KI and *Stra8* [ΔLXCXE]-*3FH* KI mice at E14.5, the cytoplasmic level of STRA8 [ΔLXCXE] seemed to be higher in germ cells at E14.5, where MEIOSIN expression was relatively weaker. Because FLAG-HA-tagged wild-type STRA8 produced from *Stra8* [wt]-*3FH* KI allele showed the same localization as endogenous STRA8, reduced nuclear retention of STRA8 [ΔLXCXE] should be due to the mutation in the LXCXE motif, indicating that nuclear localization of STRA8 was compromised without STRA8–RB binding in embryonic ovaries at E14.5.

### STRA8 is required for the timely exit from the pluripotent/primordial germ cell (PGC) state

The decision of meiotic entry precedes pre-meiotic DNA replication in mouse embryonic ovaries[4,38]. Given that STRA8 is transiently expressed during meiotic initiation in embryonic ovaries[8,39] and RB1 is expressed in ovarian germ cells[12,40], we investigated whether meiotic initiation in *Stra8* [ΔLXCXE/ΔLXCXE] germ cells was compromised when STRA8 lacking the RB-binding motif was expressed in embryonic ovaries.

To address whether *Stra8* [ΔLXCXE/ΔLXCXE] female germ cells accompanied alteration of gene expression in embryonic ovaries, we conducted scRNA-seq analyses of STRA8-positive female germ cells at E14.5, when meiotic entry was progressing. As a negative control, *Stra8*-KO female germ cells with *Stra8*-null *GFP* allele were generated by removing all the *Stra8*-coding exons from the *Stra8* [wt]-*3FH-p2A-GFP* KI allele (Supplementary Fig. 3). The STRA8-expressing cell population was isolated from the control *Stra8* [wt/wt] and *Stra8* [ΔLXCXE/ΔLXCXE] ovaries on the *Stra8-3FH-p2A-GFP* KI background by fluorescent sorting of

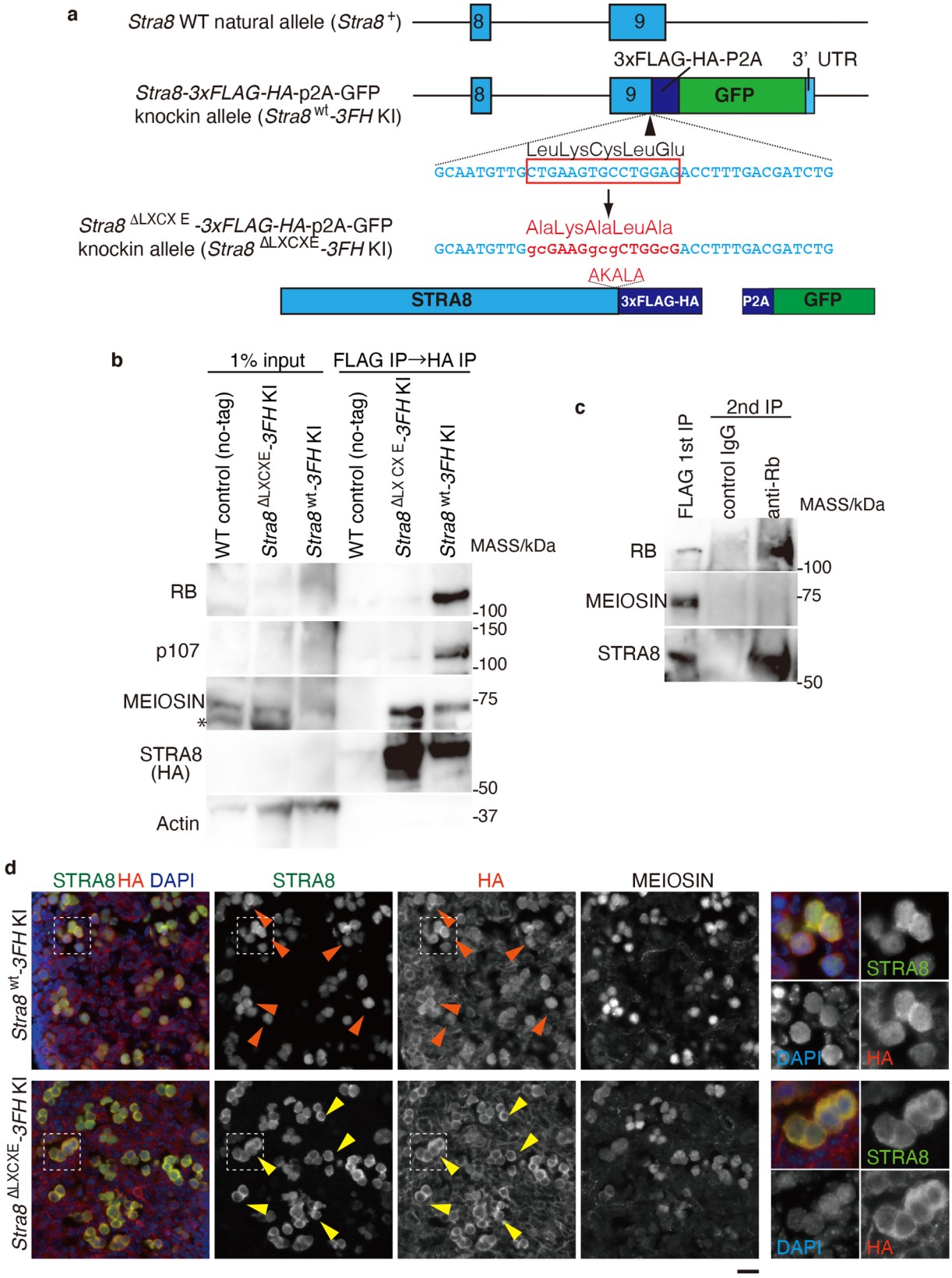

GFP-positive cells (Supplementary Fig. 4). GFP expression upon the activation of *Stra8* locus allowed us to isolate developmentally comparable *Stra8*-KO germ cells from *Stra8*-null *GFP* fetal ovaries, making it possible to comparably examine the transcriptomes of the female germ cells from *Stra8* ᵂᵗ/ᵂᵗ, *Stra8* ᴬᴸˣᶜˣᴱ/ᴬᴸˣᶜˣᴱ and *Stra8*-null E14.5 embryos (Supplementary Fig. 4a).

Intriguingly, the uniform manifold approximation projection (UMAP) of the scRNA-seq dataset indicated that gene expression patterns of single cells isolated from *Stra8* ᵂᵗ/ᵂᵗ, *Stra8* ᴬᴸˣᶜˣᴱ/ᴬᴸˣᶜˣᴱ and *Stra8*-null populations were separated into different clusters (Fig. 2a; Supplementary Fig. 5). RNA velocity analysis[41] indicated that the transcriptome of *Stra8*-null germ cells was isolated from those of

**Fig. 1 | STRA8 interacts with RB family proteins via the LXCXE motif.**
**a** Schematic illustrations of the *Stra8* wild type (WT) natural allele (*Stra8* [+]), the WT
*Stra8-3xFLAG-HA-p2A-GFP* KI allele (*Stra8* [wt]-*3FH* KI) and the LXCXE mutant *Stra8-
3xFLAG-HA-p2A-GFP* KI allele (*Stra8* [ΔLXCXE]-*3FH* KI). Blue boxes represent exons.
Coding exon 9 is followed by *3xFLAG-HA-P2A-GFP* and the 3′UTR. Triangle: CRISPR
gRNA target site. The gRNA and ssODN sequences are shown. Ala substitutions
were introduced in the LXCXE motif of STRA8. **b** Western blot of immunopreci-
pitates after tandem affinity purifications using anti-FLAG and anti-HA antibodies
from chromatin extracts of WT (non-tagged control), *Stra8* [ΔLXCXE]-*3FH* KI, and *Stra8*
[wt]-*3FH* KI mouse testes. For the negative control IP, testis extracts from WT males
that do not possess 3xFLAG-HA tagged version of STRA8 (non-tagged control) were
used. The 3xFLAG-HA-STRA8 was detected by HA antibody. *non-specific or
degraded band. Two technically independent experiments were repeated and
showed similar results. **c** The IP product of anti-FLAG antibody from *Stra8* [wt] -*3FH* KI
testis was subjected to the second IP by anti-RB1 antibody. **d** Sections from E14.5
*Stra8* [wt/wt] -*3FH* KI and *Stra8* [ΔLXCXE/+] -*3FH* KI heterozygous ovaries were immunos-
tained for DAPI, HA, and STRA8. Anti-HA antibody detected FLAG-HA-tagged wild
type STRA8 and *Stra8* [ΔLXCXE], produced from *Stra8* [wt] -*3FH* KI allele and *Stra8* [ΔLXCXE]-
*3FH* KI allele, respectively. Anti-STRA8 antibody detected both endogenous wild
type STRA8 and FLAG-HA-tagged STRA8/STRA8[ΔLXCXE]. While FLAG-HA-tagged
STRA8[ΔLXCXE] was expressed at the same timing as wild type STRA8, its localization
was skewed to the cytoplasm. Enlarged images are shown on the right. Red
arrowhead: nuclear dominant localization of STRA8. The technically independent
experiments were repeated more than three times. Yellow arrowhead: cytoplasmic
dominant localization of STRA8[ΔLXCXE]. Scale bar: 20 μm. See also Supplementary
Figs. 1 and 2.

*Stra8* [wt/wt] and *Stra8* [ΔLXCXE/ΔLXCXE] rather than converging to them (Fig. 2b).
Consistent with a previous study[18], gene enrichment analysis showed
that the genes related to stem cell maintenance, such as *Oct4/Pou5f1*,
*Nanog*, *Sox2*, *Esrrb*, and *Klf5*, were highly expressed in *Stra8*-null germ
cells (Fig. 2c, d; Supplementary Data 1). Indeed, the expression of
NANOG protein persisted in *Stra8*-null ovaries at E14.5 (Fig. 2e). Thus,
*Stra8*-null germ cells were yet to exit from the pluripotent state at E14.5.
This phenomenon was, at least in part, consistent with previous reci-
procal observations in polycomb repressive complex1 (PRC1) KO
showing precocious upregulation of STRA8 in XX germ cells, leading to
precocious repression of pluripotent genes and concomitant upregu-
lation of a subset of meiotic genes as early as at E12.5[42]. Furthermore,
PGC marker genes, *Prdm14* and *Nanos3*[43,44], also showed higher expres-
sion in *Stra8* [ΔLXCXE/ΔLXCXE] and *Stra8*-null oogonia compared to *Stra8* [wt/wt]
(Fig. 2f). BMP signaling synergistically induces the expression of a broad
range of genes through genome-wide activation by ZGLP1 prior to
meiotic initiation in female germ cells[27,28]. Consistent with a previous
study showing that expression of *Zglp1 mRNA* reached peaks at E12.5 and
E13.5 and declined afterward, and that expression of ZGLP1 started to
appear at E12.5 and precedes that of STRA8 at the protein level[27], *Zglp1*
and other downstream genes of BMP signaling (*Id-1*, *Id2*, *Id3*, *Gata2*,
*Msx1*, and *Msx2*) were expressed in *Stra8*-null oogonia, albeit at variable
levels (Fig. 2g; Supplementary Fig. 5d). These data suggest that *Stra8*-null
oogonia undergo female-specific sex determination and respond to
BMP signaling prior to meiosis. *Zglp1* expression is independent of
STRA8 and its pattern is transient, compared to other downstream
genes of BMP signaling, such as *Msx1*[28]. We found that *Zglp1* expression
remained relatively high in the *Stra8*-null oogonia, compared with WT,
and that *Zglp1* expression was intermediate in the *Stra8* [ΔLXCXE /ΔLXCXE]
mutant oogonia. This suggests that the developmental progression of
*Stra8* [ΔLXCXE/ΔLXCXE] oogonia is intermediate between WT and *Stra8*-null
mice. RA-responsive genes, such as *Rec8*[22], were also expressed in *Stra8*-
null oogonia, indicative of the response to RA (Fig. 2h; Supplementary
Fig. 5d). Altogether, STRA8 is required for the timely exit from plur-
ipotent/PGC status prior to meiosis in female germ cells.

## STRA8−RB interaction is required for the timely progression into pre-meiotic S phase in female mice

Gene expression patterns of single cells isolated from *Stra8* [wt/wt], *Stra8*
[ΔLXCXE /ΔLXCXE] and *Stra8*-null mice were separated into 11 clusters at E14.5
(Supplementary Fig. 5a–c; Supplementary Data 2). Since RB1 plays a
key role in the repression of S phase progression, we asked whether
the alteration of transcriptome in *Stra8* [ΔLXCXE /ΔLXCXE] versus *Stra8* [wt/wt]
accompanied the change in cell cycle status at E14.5. Cell cycle
scoring analysis of the scRNA-seq dataset approximately estimates, if not
exactly specifies, the cell cycle stages from the genes expressed
according to the cell cycle status (Fig. 3a–c; Supplementary Data 3). We
defined the G1/S phase transition in the GFP-positive germ cells by the
expression patterns of cell cycle-associated reference genes (Fig. 3c),
so that the alteration of cell cycle genes can be assessed separately
from developmentary-regulated gene expression.

G1/S phase was overall reduced in *Stra8*-null compared to *Stra8* [wt/wt]
at E14.5 (Fig. 3a, c), suggesting that STRA8 had an impact on S
phase progression at this time point, which was consistent with the
evidence of STRA8 being required for the decision to enter meiosis
before pre-meiotic DNA replication[4]. We noticed a STRA8-expressing
subpopulation that was present in *Stra8* [wt/wt], but largely missing in
*Stra8* [ΔLXCXE /ΔLXCXE] mice at E14.5 (shown by dashed oval corresponding
to clusters 5 and 7 in Fig. 4a), corresponding to the terminal
direction estimated by RNA velocity analysis (Fig. 2b). Thus, develop-
mental progression of STRA8-expressing germ cells was delayed in
*Stra8* [ΔLXCXE /ΔLXCXE] mice compared to *Stra8* [wt/wt]. Strikingly, cell cycle
scoring analysis estimated that clusters 5 and 7 overall corresponded
to a feature of G1/S phase transition, which was markedly reduced
in *Stra8* [ΔLXCXE/ΔLXCXE] (Fig. 3b, c). Consistently, our immunostaining con-
firmed that the STRA8-expressing population that was labeled by EdU
was significantly reduced in *Stra8* [ΔLXCXE/ΔLXCXE] compared to the control at
E14.5 (Fig. 3d), implying that progression into S phase was compro-
mised in *Stra8* [ΔLXCXE /ΔLXCXE] germ cells. Therefore, the LXCXE motif of
STRA8 affects G1/S phase progression.

Crucially, while *Rb1* was expressed at a comparable level both in
*Stra8* [ΔLXCXE/ΔLXCXE] and control, albeit the expression of *p107* was less in
*Stra8* [ΔLXCXE/ΔLXCXE], (Fig. 3e), the overall expression levels of E2F-target
genes were higher in the G1/S population (corresponding to clusters 5
and 7 in Supplementary Fig. 5a; Fig. 3f), which was largely missing in
*Stra8* [ΔLXCXE /ΔLXCXE] germ cells at E14.5 (Supplementary Fig. 5b). There-
fore, in female meiotic initiation, the activation of E2F-responsive
genes is controlled by STRA8−RB interaction. In the canonical cell
cycle, dissociation of RB from E2F by cyclin-dependent kinase (CDK)4/
6-mediated phosphorylation of RB leads to consequent activation of
E2F-target genes[30]. However, it should be mentioned that *Cdkn2a*
encoding alternative splicing products, p16[INK4a] and p19[ARF][45,46], was
upregulated in the G1/S population in STRA8-expressing germ cells
(clusters 3, 5, and 7; Fig. 3g), which may indirectly lead to decreased
CDK activity[47]. Since *Cdkn2a* locus is shared by alternative splicing
products p16[INK4a] and p19[ARF], and 10× scRNA-seq acquires sequence
reads derived mostly from the 3′ side of mRNA, further investigation is
required to determine whether either p16[INK4a] or p19[ARF] contributes to
CDK inhibition in this context.

Viral oncoproteins, such as human papilloma virus E7, SV40 large
T antigen, and adenovirus E1A, bind to RB through the LXCXE motif
and inactivate RB to accelerate S phase progression[48,49]. Therefore,
STRA8 may sequester RB from E2F in female germ cells and promote
the activation of S phase-related genes (Fig. 3h), which is similar to the
mechanism executed by viral oncoproteins.

We noticed that cluster 6 (Supplementary Fig. 5a) also repre-
sented a feature of the G1/S transition, which was present in both the
control and *Stra8* [ΔLXCXE /ΔLXCXE] germ cells (Fig. 3b). This population
presumably represents STRA8-positive germ cells that were still under
the mitotic cell cycle, accompanied by upregulation of E2F-target
genes, because activation of meiotic genes was yet to be shown in this
cluster (see also discussion below; Fig. 4a).

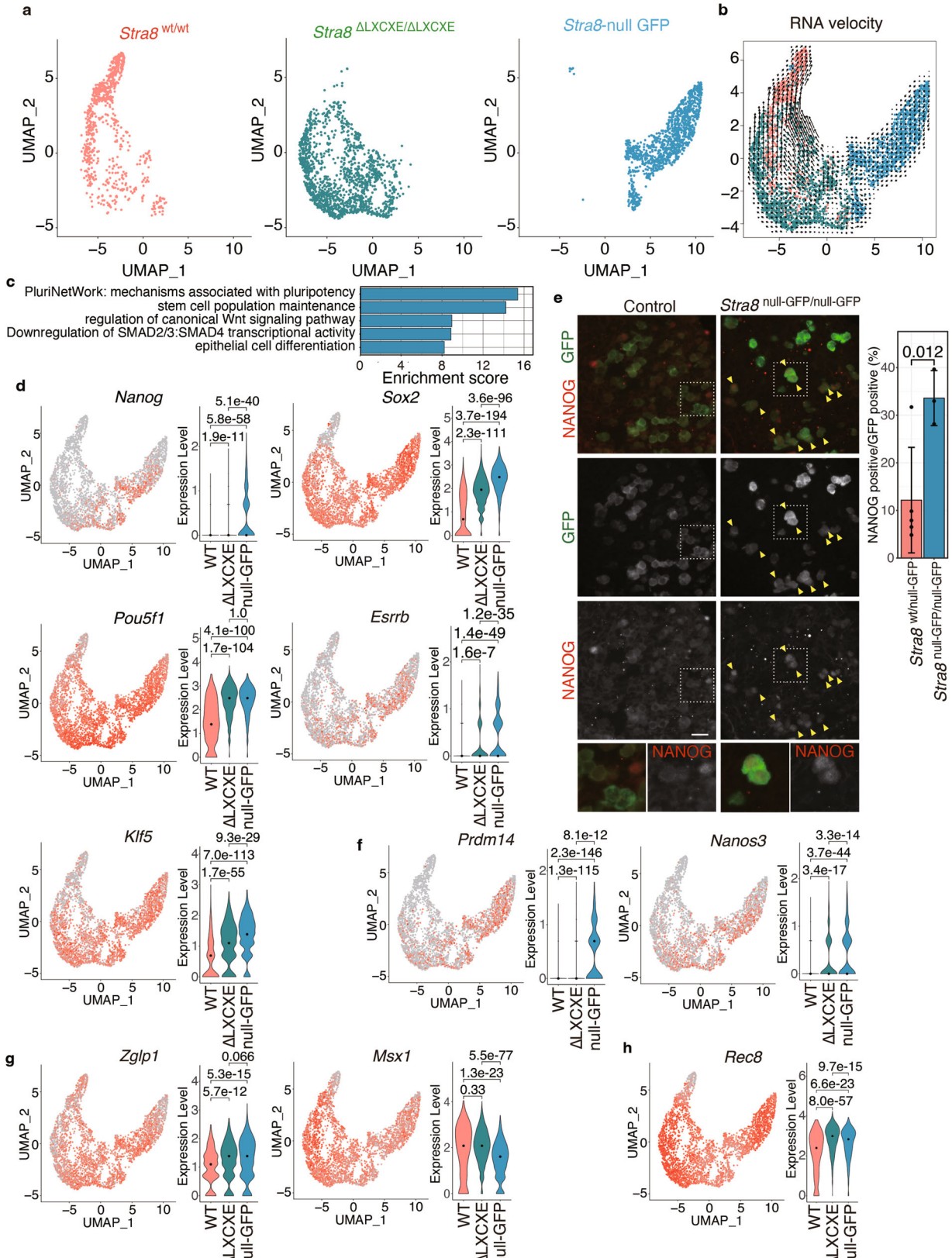

## STRA8–RB interaction is required for the timely meiotic initiation in female mice

RNA velocity analysis estimated that STRA8-expressing germ cells progressed along the trajectory toward two distinct directions, cluster 5 or 6, at E14.5 (Figs. 4a, 2b). Gene Enrichment analysis of differentially expressed genes at E14.5 revealed that clusters 5 and 7 were associated with meiosis-related gene expression (Fig. 4b; Supplementary Fig. 5a; Supplementary Data 2). Because clusters 5 and 7 coincided with a feature of the G1/S phase (Fig. 3a) and accompanied the upregulation of meiosis-related genes (Fig. 4b), these clusters likely represent pre-meiotic G1/S phase transition, or early meiotic prophase at most, at E14.5. Cluster 5 accompanied *Meiosin* expression at the highest level

**Fig. 2 | Exit from pluripotent state was compromised in *Stra8* ^ΔLXCXE /ΔLXCXE^ embryonic germ cells. a** UMAP representation of scRNA-seq transcriptome profiles for GFP-expressing germ cells from E14.5 *Stra8* ^wt/wt^-*3xFH-P2A-GFP*, *Stra8* ^ΔLXCXE /ΔLXCXE^ -*3xFH-P2A-GFP* and *Stra8* ^null GFP/ null GFP^ knock-in embryonic ovaries. **b** The developmental trend of the individual cells was estimated by RNA velocity analysis. The velocities are visualized on the merged UMAP plot. **c** Gene ontology analysis of DEGs in the *Stra8* ^null GFP^ germ cells. See also Supplementary Data 1. **d** Expression patterns of the pluripotent genes *Pou5f1*, *Nanog*, *Sox2*, *Esrrb*, and *Klf5* on the UMAP plot (left) and violin plot (right). *p*-values were indicated in each plot (two-sided wilcoxon rank-sum test). Black dot indicates median. **e** Sections from E14.5 *Stra8* ^null^ ^GFP^ homozygous and heterozygous control ovaries were stained as indicated. Scale bar: 20 μm. Yellow arrowhead: NANOG positive cells. Enlarged images are shown on the bottom. Quantification of NANOG-positive cells per total number of GFP-expressing germ cells in the control *Stra8* ^wt/null GFP^ (*n* = 5) and *Stra8* ^null GFP/null GFP^ (*n* = 3) ovaries are shown on the right (bar plot of mean values with +/− SD). *p* = 0.01158 (two-sided welch *t*-test). **f** Expression patterns of PGC marker genes *Prdm14* and *Nanos3* are shown as in (**d**). **g** Expression patterns of BMP responsive marker genes *Zglp1* and *Id1* are shown as in (**d**). (**h**) Expression pattern of RA responsive marker gene *Rec8* is shown as in (**d**). WT: *Stra8* ^wt/wt^, ΔLXCXE: *Stra8* ^ΔLXCXE^, null-GFP: *Stra8* ^null-GFP^. See also Supplementary Figs. 3, 4, 5, Supplementary Data 1.

among the STRA8-expressing subclusters at E14.5 (Fig. 4c), implying that *Meiosin* expression coincided with G1/S transition in female germ cells. Since most germ cells in *Stra8* ^ΔLXCXE /ΔLXCXE^ were yet to reach the state represented by clusters 5 and 7, *Meiosin* expression was delayed and underrepresented in *Stra8* ^ΔLXCXE /ΔLXCXE^ at E14.5.

Cluster 6 also represented a feature of G1/S transition (Fig. 3a). However, since the Cluster 6 population was yet to express meiotic genes, it presumably represented a STRA8-expressing population that was still under the mitotic cell cycle.

Notably, the expression of meiotic genes associated with clusters 5 and 7 was underrepresented in *Stra8* ^ΔLXCXE /ΔLXCXE^ at E14.5 (Fig. 4d). As shown by MEIOSIN being less expressed at the protein level in *Stra8* ^ΔLXCXE /ΔLXCXE^ (Fig. 4e), meiotic gene activation was compromised in the absence of STRA8–RB interaction at E14.5. Altogether, these observations suggest that STRA8–RB interaction is a prerequisite for timely meiosis initiation in female germ cells, so that meiotic gene activation coincides with the G1/S transition of the cell cycle (Fig. 4f).

## Progression of meiotic prophase is delayed in the absence of STRA8–RB in female germ cells

To delineate the fate of *Stra8* ^ΔLXCXE /ΔLXCXE^ germ cells, we further conducted scRNA-seq analyses of the germ cells at E15.5 (Fig. 5a; Supplementary Fig. 6). Our initial attempt to collect STRA8-GFP-expressing germ cells at E15.5, was technically difficult due to the insufficient number of starting cells for scRNA-seq library preparation. For this purpose, we isolated DDX4-positive germ cells at E15.5 from the control and *Stra8* ^ΔLXCXE /ΔLXCXE^ ovaries on the *Ddx4-Venus*-background by fluorescent sorting of VENUS-positive cells (Supplementary Fig. 4b).

Gene expression patterns of single cells isolated from *Stra8* ^ΔLXCXE /ΔLXCXE^ and the littermate control *Stra8* ^ΔLXCXE/+^ mice were separated into 10 Clusters at E15.5 (Fig. 5b; Supplementary Fig. 6a; Supplementary Data 4). Notably, cell cycle scoring analysis estimated that clusters 0, 4, and 6 exhibited a feature of G1/S transition and were present in the *Stra8* ^ΔLXCXE /ΔLXCXE^ germ cell population (Fig. 5c; Supplementary Fig. 6a). Consistently, EdU incorporation in *Stra8* ^ΔLXCXE /ΔLXCXE^ germ cells was increased at E15.5 compared to E14.5 (Fig. 5d, compare with Fig. 2c), suggesting that *Stra8* ^ΔLXCXE /ΔLXCXE^ germ cells eventually entered the G1/ S phase transition at E15.5.

Furthermore, gene enrichment analysis showed that clusters 0, 4, and 6 were associated with the expression of meiosis-related genes and were present in the *Stra8* ^ΔLXCXE /ΔLXCXE^ germ cell population (Supplementary Fig. 6b–d; Supplementary Data 4). Accordingly, clusters 0, 4, and 6 were accompanied by the expression of *Meiosin* and meiotic genes at E15.5 (Fig. 5e, h). Indeed, MEIOSIN-positive cells appeared one day later on E15.5 in *Stra8* ^ΔLXCXE /ΔLXCXE^ ovaries (Fig. 5f), although it had been less present at E14.5 (Fig. 4e). Concomitantly with MEIOSIN expression, the nuclear localization of FLAG-HA-tagged STRA8 ^ΔLXCXE^ partially, if not fully, resumed at E15.5. This may be partly due to the recruitment of STRA8 ^ΔLXCXE^ into the nucleus via its interaction with MEIOSIN (Fig. 5g), analogically suggested by the nuclear localization of STRA8 depending on MEIOSIN in pre-leptotene spermatocytes (Supplementary Fig. 2a, b). Since the nuclear/cytoplasmic localization ratio of STRA8 was lower in *Stra8* ^ΔLXCXE /ΔLXCXE^ cells than in the control, nuclear localization of STRA8 depends on both RB and MEIOSIN in

female germ cells. It should be mentioned that the control and *Stra8* ^ΔLXCXE/ΔLXCXE^ cells transitioned through different trajectories, converging to cluster 0 (Fig. 5b). This implies that delayed progression into the G1/ S phase was accompanied by an overall alteration in gene expression under the loss of STRA8–RB interaction. These data suggest that *Stra8* ^ΔLXCXE /ΔLXCXE^ germ cells eventually reach the G1/S phase transition and activate the meiotic genes at E15.5. Thus, we concluded that S phase entry and meiotic gene activation were delayed in the absence of STRA8–RB interaction.

Consistently, *Stra8* ^ΔLXCXE /ΔLXCXE^ oocytes showed homolog synapsis and progressed through the meiotic prophase with an accompanying delay, as assessed by immunostaining of SYCP1 at E15.5, E16.5, and E18.5, respectively (Fig. 6a). Cytological analysis of spread chromosomes demonstrated that transition of the number of DMC1 foci (a marker of ssDNA in DBS site) across the stages of meiotic prophase was comparable between the control and *Stra8* ^ΔLXCXE /ΔLXCXE^ oocytes, suggesting that generation of DSBs, and meiotic recombination were proficient in *Stra8* ^ΔLXCXE /ΔLXCXE^ oocytes, if not all (Fig. 6b). Furthermore, the foci of MLH1, a marker of crossover recombination, were observed in *Stra8* ^ΔLXCXE /ΔLXCXE^ pachytene oocytes (Fig. 6c). Although slight decrease in the number of MLH1 foci may be due to delayed progression of meiotic prophase, crossover recombination seems to be proficient in *Stra8* ^ΔLXCXE /ΔLXCXE^ oocytes (see discussion below and Fig. 7c). These data suggest that meiotic entry and subsequent meiotic prophase were delayed by approximately 1 day in *Stra8* ^ΔLXCXE /ΔLXCXE^ germ cells. Collectively, these results suggest that the STRA8–RB interaction is required for pre-meiotic S phase entry and meiotic gene activation, so that the meiotic program is installed in the S phase for timely meiotic entry in female germ cells.

## Oocyte development is delayed in *Stra8* ^ΔLXCXE /ΔLXCXE^ ovary

In females, upon reaching the diplotene stage around birth, oocytes undergo dictyate/diplotene arrest in the middle of the meiotic prophase, which is equivalent to long-term G2 arrest of the cell cycle. From postnatal day 2 (PD2) onward, early oocyte maturation is triggered after activation of primordial follicles, which accompanies dynamic gene expression changes as primordial follicles transition to primary follicles (primordial-to-primary-follicle transition: PPT)[50]. We investigated the fate of *Stra8* ^ΔLXCXE /ΔLXCXE^ oocytes that had undergone delayed meiotic prophase. To delineate whether *Stra8* ^ΔLXCXE /ΔLXCXE^ oocytes displayed alterations in gene expression after having undergone the delayed meiotic prophase, we further conducted scRNA-seq analyses of oocytes at E18.5. For this purpose, we isolated oocytes at E18.5, from the control and *Stra8* ^ΔLXCXE /ΔLXCXE^ ovaries on the *Ddx4-Venus*-background, by fluorescent sorting of VENUS-positive cells (Fig. 7; Supplementary Fig. 7).

UMAP of the scRNA-seq dataset showed that the overall gene expression profiles were different between the control and *Stra8* ^ΔLXCXE /ΔLXCXE^ oocytes at E18.5 (Supplementary Fig. 7a). RNA velocity indicated that the control oocytes underwent development along a trajectory toward clusters 0 and 5, which were largely missing in *Stra8* ^ΔLXCXE /ΔLXCXE^ oocytes at E18.5 (Supplementary Fig. 7b–d). Gene enrichment analyses revealed that cluster 0 represented the genes that are related to oogenesis (Supplementary Fig. 7e; Supplementary Data 5). Instead, the gene expression profiles of *Stra8* ^ΔLXCXE /ΔLXCXE^

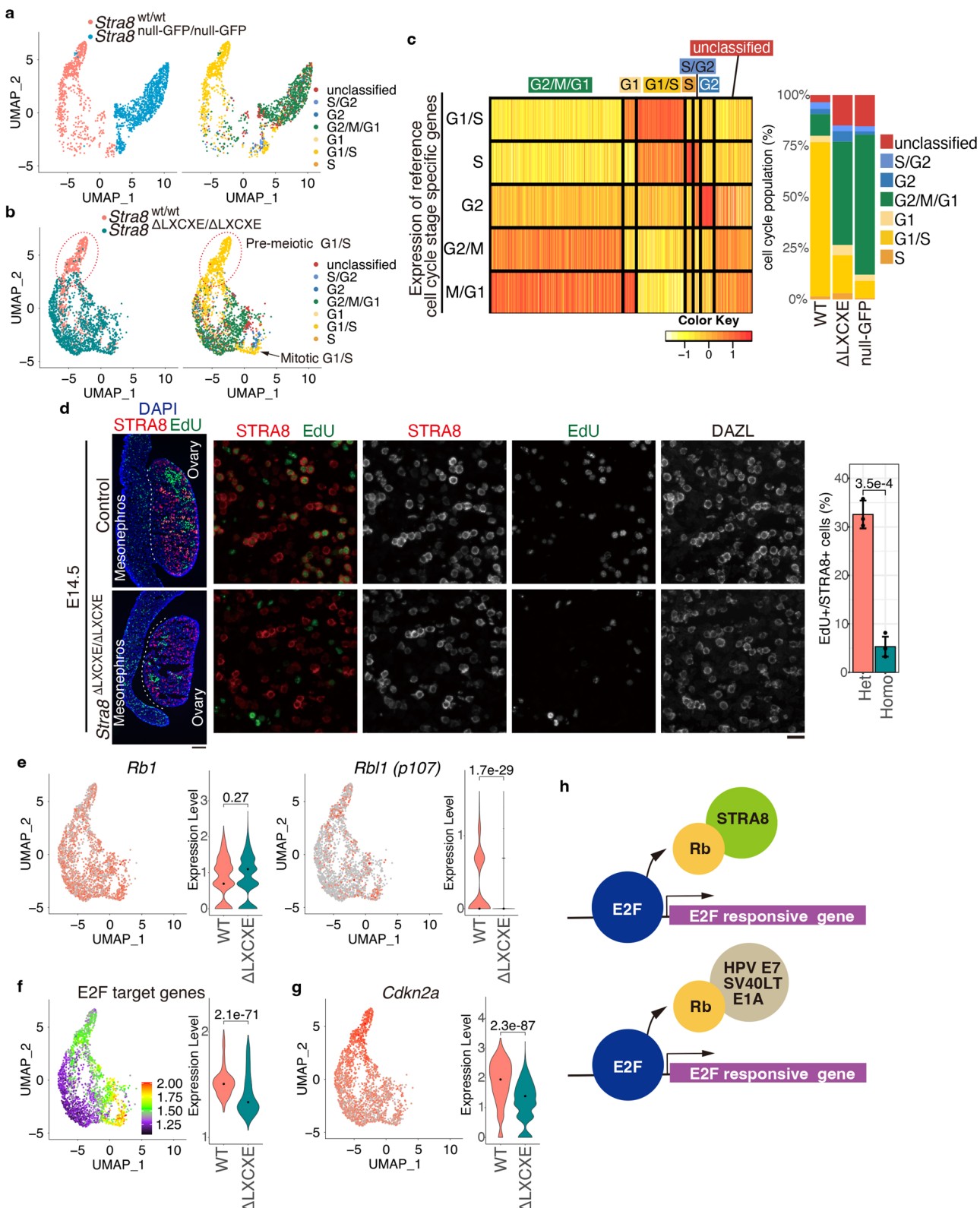

oocytes transitioned along two different trajectories toward either cluster 6 or 8 that were absent in the control oocyte population at E18.5 (Supplementary Fig. 7c, d).

To compare the overall developmental progression of *Stra8* ΔLXCXE/ΔLXCXE oocytes to the control, we conducted a combined analysis of scRNA-seq data of germ cells from E14.5, E15.5, and E18.5 ovaries (Fig. 7a). *Stra8* ΔLXCXE/ΔLXCXE oocytes hardly showed the expression of the genes that encode key factors for follicle maturation/PPT, such as

*Nobox* or *Lhx8*, suggesting that developmental progression of *Stra8* ΔLXCXE /ΔLXCXE oocytes was overall delayed or arrested at E18.5 as a consequence of delayed entry into meiosis and subsequent delayed progression of meiotic prophase (Fig. 7b). Furthermore, gene enrichment analyses revealed that cluster 8 of *Stra8* ΔLXCXE/ΔLXCXE oocyte population indicated the upregulation of genes related to p53-dependent damage response, such as *Cdkn1a*, *Gadd45b*, *Bbc3*, and *Pmaip1*[51] (Supplementary Fig. 7e, f; Supplementary Data 5), suggesting that p53 levels were

**Fig. 3 | S phase progression was compromised in *Stra8* $^{\Delta LXCXE /\Delta LXCXE}$ female germ cells. a** Merged UMAP representation is shown for scRNA-seq transcriptomes of STRA8-expressing germ cells from *Stra8* $^{wt/wt}$ and stage-matched germ cells from *Stra8* $^{null\,GFP/\,null\,GFP}$ ovaries at E14.5 (left) and the estimated cell cycle status (right). Individual cells are colored as indicated. **b** Merged UMAP representation is shown for *Stra8* $^{wt/wt}$ and *Stra8* $^{\Delta LXCXE /\Delta LXCXE}$ ovaries at E14.5 and the estimated cell cycle status of them as in (**a**) Red dotted circle indicates pre-meiotic G1/S subpopulation that was largely missing in *Stra8* $^{\Delta LXCXE /\Delta LXCXE}$, corresponding to Clusters 5 and 7 in Supplementary Fig. 5a. **c** Heat map shows the expressions of cell cycle-associated genes in each cell (left). Proportion of UMAP-defined cell cycle status are shown (right). WT: *Stra8* $^{wt/wt}$, ΔLXCXE: *Stra8* $^{\Delta LXCXE/\Delta LXCXE}$, null GFP: STRA8-null germ cells. **d** Sections from *Stra8* $^{\Delta LXCXE /\Delta LXCXE}$ and control ovaries at E14.5 were stained as indicated. Scale bars: 100 μm for whole sections (left); 20 μm (middle). DAZL represents a marker of germ cells. Quantification of EdU-incorporated cells per total number of STRA8-expressing germ cells in the control (*n* = 3) and *Stra8* $^{\Delta LXCXE}$ $^{/\Delta LXCXE}$ (*n* = 3) ovaries are shown on the right (bar plot of mean values with +/− SD). *p* = 0.000349 (two-sided welch *t*-test). **e** Expression patterns and levels shown by UMAP plot and violin plot, respectively, for *Rb1*(left) and *Rbl1* (*p107*) (right). n.s.: not significant. *p* = 0.268 for *Rb1* and *p* = 1.75 × 10$^{-29}$ for *Rbl1* (two-sided wilcoxon rank-sum test). Black dots indicate median. WT: *Stra8* $^{wt/wt}$, ΔLXCXE: homozygous *Stra8* $^{\Delta LXCXE/\Delta LXCXE}$. **f** The overall expression pattern of E2F-target genes on the UMAP plot and violin plot. *p* = 2.1 × 10$^{-71}$ (two-sided Wilcoxon rank-sum test). Black dots indicate median. **g** The overall expression pattern of *Cdkn2a* on the UMAP plot. *p* = 2.3 × 10$^{-87}$ (two-sided Wilcoxon rank-sum test). Black dots indicate median. **h** Schematic illustrations of RB (RB1 and p107) dissociation from E2F and de-repression of E2F-target genes by RB binding to STRA8 (upper), and RB binding to viral oncoproteins (HPV E7, SV40LT, Adenovirus E1A) (lower). See also Supplementary Figs. 4, 5, Supplementary Data 2 and 3.

elevated in *Stra8* $^{\Delta LXCXE /\Delta LXCXE}$ oocytes at E18.5. Indeed, TdT-mediated dUTP nick-end labeling (TUNEL)-positive oocytes were increased in *Stra8* $^{\Delta LXCXE/\Delta LXCXE}$ ovaries from E18.5 to PD0, suggesting that the mutant oocytes were eliminated by apoptosis (Fig. 7c).

Given that *Stra8* $^{\Delta LXCXE /\Delta LXCXE}$ oocytes exhibited delayed developmental progression and some exhibited apoptosis around birth, we investigated the consequence in the *Stra8* $^{\Delta LXCXE /\Delta LXCXE}$ ovary at a later time point. In contrast to the male phenotype (Supplementary Fig. 2), *Stra8* $^{\Delta LXCXE /\Delta LXCXE}$ females showed infertility over a period of 6 months (*n* = 9). Histological analysis revealed that ovaries were degenerated with no follicles at the age of eight-week-old in *Stra* $^{\Delta LXCXE/\Delta LXCXE}$ similar to *Stra8* KO (Fig. 8a)[38]. Notably, primordial follicles had already been lost in *Stra8* $^{\Delta LXCXE /\Delta LXCXE}$ ovaries as early as postnatal day17 (Fig. 8b), indicating that oocyte loss was accelerated in *Stra8* $^{\Delta LXCXE /\Delta LXCXE}$ ovaries. Furthermore, the numbers of transition, primary, secondary, and antral follicles were significantly decreased in *Stra8* $^{\Delta LXCXE /\Delta LXCXE}$ ovaries compared to those in age-matched controls at these time points. These results suggest that premature oocyte depletion occurred in RB-binding-deficient *Stra8* mutant females, at least by P17.

To examine cellular morphology, we isolated a small number of residual oocytes from *Stra8* $^{\Delta LXCXE /\Delta LXCXE}$ ovaries (3-week-old) after hormonal injection. *Stra8* $^{\Delta LXCXE /\Delta LXCXE}$ oocytes exhibited germinal vesicles (GV) and zona pellucida, which had the same morphology as wild-type GV oocytes. We next examined the chromosome morphologies of *Stra8* $^{\Delta LXCXE /\Delta LXCXE}$ oocytes that were collected 4 h after germinal vesicle break down (GVBD), when wild-type chromosomes would undergo condensation before the first meiotic division. Crucially, Giemsa staining of the chromosomes revealed that 20 pairs of bivalents with chiasmata were observed in *Stra8* $^{\Delta LXCXE /\Delta LXCXE}$ oocytes, as in wild-type oocytes (Fig. 8c). In contrast, 40 univalents were observed in *Stra8* KO oocyte-like cells (Fig. 8c), which was similar to *Meiosin* KO oocyte-like cells[9]. Whereas the presence of chiasmata in *Stra8* $^{\Delta LXCXE /\Delta LXCXE}$ oocytes indicated that they had progressed through the meiotic prophase and had experienced meiotic recombination (Fig. 6), the lack of chiasmata in *Stra8* KO oocyte-like cells indicated that they had not accomplished these phases[38]. Therefore, premature oocyte loss observed in *Stra8* $^{\Delta LXCXE /\Delta LXCXE}$ and *Stra8* KO females was derived from different primary defects in meiotic prophase, although those ovaries exhibited histologically similar phenotypes. Altogether, delayed entry into meiosis consequently led to premature loss of the oocyte pool as a consequence in *Stra8* $^{\Delta LXCXE /\Delta LXCXE}$ ovaries.

## Discussion
Our genetic study combined with single-cell transcriptome analyses of STRA8-expressing germ cells unraveled a previously unforeseen female-specific mechanism of meiotic initiation. We showed that STRA8 binds to RB1 and p107 via the LXCXE motif, forming a subcomplex distinct from the STRA8-MEIOSIN complex in the testis (Fig. 1). Due to the limited availability of materials, it was technically difficult to show the interaction between STRA8 and RB by IP-MS experiment using extracts from the embryonic ovary although we tried to do it. Although *Stra8* $^{\Delta LXCXE /\Delta LXCXE}$ and *Stra8* KO ovaries exhibited histologically similar phenotypes (Fig. 8), they bore different primary defects. Whereas female germ cells failed to undergo meiosis in *Stra8* KO (Fig. 2), those in *Stra8* $^{\Delta LXCXE /\Delta LXCXE}$ ovaries showed delayed progression into G1/S transition (Fig. 3a–c), and delayed entry into meiosis (Fig. 5). These results indicate that the STRA8–RB1 interaction plays a specific role in meiotic initiation in female germ cells.

Since the activation of E2F responsive genes and subsequent progression into S phase were delayed in *Stra8* $^{\Delta LXCXE /\Delta LXCXE}$ oogonia (Fig. 5), STRA8–RB1 interaction is required for scheduled progression into pre-meiotic S phase in females. Crucially, the STRA8–RB1 interaction controls the activation of E2F target genes in female germ cells (Fig. 3e). Therefore, we propose that STRA8 sequesters RB1 from E2F to derepress the expression of E2F-responsive genes (Fig. 3h), which is similar to the mechanism known in viral oncoproteins[48,49]. Similarly, it is possible that STRA8 sequesters p107 from the E2F4/5 containing-DREAM complex, leading to disassembly of the repressive form[32]. Since the p107-bound DREAM complex represses both G1/S and G2/M cell cycle-related genes through the DNA-binding specificities of E2F4/5 (E2F responsive element) and MuvB (CHR element)[32], it is likely that the STRA8-p107 interaction may contribute to the de-repression of CHR-mediated genes for cell cycle progression.

Subsequently, STRA8, in collaboration with MEIOSIN, synchronized meiotic gene activation at the G1/S transition in female germ cells (Fig. 4f). The mechanisms executed by the STRA8–RB and STRA8-MEIOSIN subcomplexes coordinate the synchronization of S phase progression and meiotic gene activation for timely meiotic entry in female germ cells, so that the meiotic gene expression program is installed in the S phase. This idea is consistent with a previous observation that the ectopic expression of STRA8 leads to an apparent sign of meiotic entry even in XY germ cells; otherwise, they should remain in G0/G1 arrest[15,52]. In contrast, STRA8–RB interaction is dispensable for meiotic initiation in male (Supplementary Fig. 2). This could presumably be because de-repression of E2F responsive genes is mediated by RB phosphorylation at the pre-meiotic S phase in males. Alternatively, STRA8 may amplify the expressions of E2F responsive genes in pre-leptotene spermatocytes, as previously proposed[10]. These lines of evidence account for the requirement of STRA8–RB interaction for meiosis entry in mouse females.

The delayed meiotic entry consequently led to p53-dependent damage response, and premature elimination of the oocyte pool in *Stra8* $^{\Delta LXCXE/\Delta LXCXE}$ ovaries (Fig. 7c, Supplementary Fig. 7e, f). Although homolog synapsis, meiotic recombination and crossover were apparently proficient in *Stra8* $^{\Delta LXCXE /\Delta LXCXE}$ oocytes (Fig. 6), we cannot formally exclude a possibility that p53-dependent damage response might in part derive from a hidden defect in these processes. Alternatively, it is possible that altered gene expression patterns as a result of loss of STRA8-RB1/p107 interaction might trigger

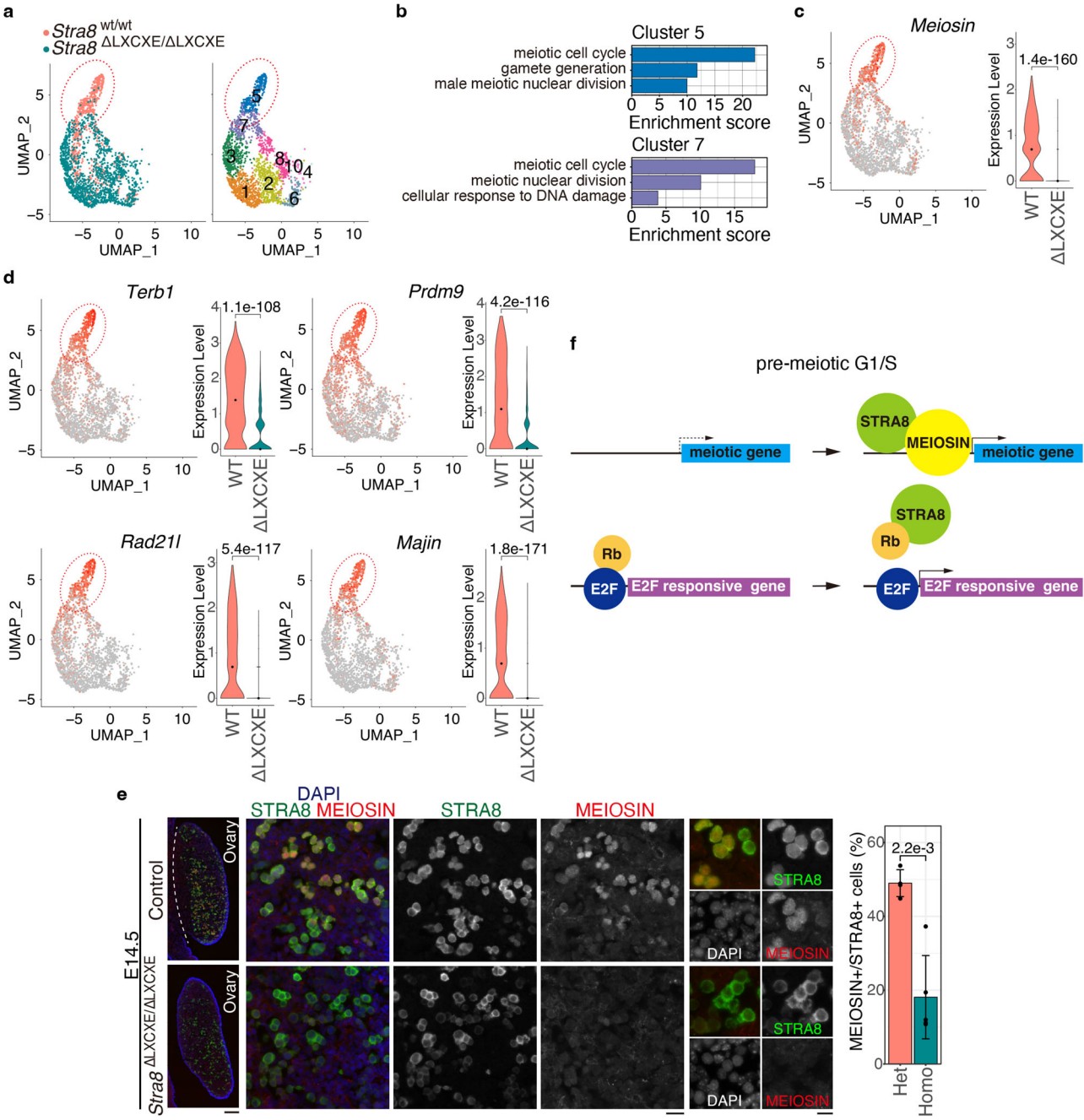

**Fig. 4 | Timely meiotic entry was compromised in the absence of STRA8-RB interaction in embryonic ovary at E14.5. a** Merged UMAP representation is shown for STRA8-expressing germ cells from E14.5 *Stra8*<sup>wt/wt</sup> and *Stra8*<sup>ΔLXCXE/ΔLXCXE</sup> mice (left). Clustering analysis of different gene expression patterns on UMAP-defined scRNA-seq transcriptomes is shown (right). See also Supplementary Fig. 5. Red dotted oval corresponds to pre-meiotic G1/S subpopulation that was largely missing in *Stra8*<sup>ΔLXCXE/ΔLXCXE</sup> in Fig. 4A. Grey arrows indicate developmental directions estimated by RNA velocity (see Fig. 2b). **b** Gene enrichment analysis of DEGs in the UMAP-defined cell clusters. The bar graph shows enrichment score -log10 (*p*-value) for each term. See also Supplementary Fig. 5c for other clusters. **c** Expression pattern and level of *Meiosin* are shown on the UMAP plot and violin plot, respectively. n.s.: not significant. $p = 1.38 \times 10^{-160}$ (two-sided wilcoxon rank-sum test). Black

dots indicate median. WT: *Stra8*<sup>wt/wt</sup>, ΔLXCXE: homozygous *Stra8*<sup>ΔLXCXE/ΔLXCXE</sup>. **d** Expression pattern and level of meiosis-related genes are shown on the UMAP plot and violin plot, respectively. n.s.: not significant. *p*-values were indicated in each plot (two-sided wilcoxon rank-sum test). Black dots indicate median. **e** Sections from *Stra8*<sup>ΔLXCXE/ΔLXCXE</sup> and control ovaries at E14.5 were stained as indicated. Scale bars: 100 μm for whole sections (left); 20 μm (middle); 10 μm for enlarged images (right). Quantification of MEIOSIN-positive cells per STRA8-positive germ cells in *Stra8*<sup>ΔLXCXE/+</sup> (*n* = 4) and *Stra8*<sup>ΔLXCXE/ΔLXCXE</sup> (*n* = 5) ovaries are shown by bar plot of mean with +/− SD (the most right). $p = 2.2 \times 10^{-3}$ (two-sided welch *t*-test). **f** Schematic model of the coordinated activation of E2F-responsive genes and meiotic genes. See also Supplementary Fig. 5, Supplementary Data 2.

p53-dependent damage response. Thus, timely entry into meiosis is required for proper developmental progression and maintenance of oocyte pool in female.

In embryonic gonads, XX and XY germ cells are initially indistinguishable, and an obvious sexual difference emerges at E12.5 onward.

Concordant with sex determination, while XY germ cells undergo G0/G1 arrest by RB1, XX germ cells bypass G0/G1 arrest even in the presence of RB1 and instead enter meiosis[12,13]. In the absence of *Rb1*, XY germ cells actively proliferate at E14.5- E16.5 in ex vivo culture of fetal testes with delayed cell cycle arrest[12], and oocytes raise ovarian teratoma[40].

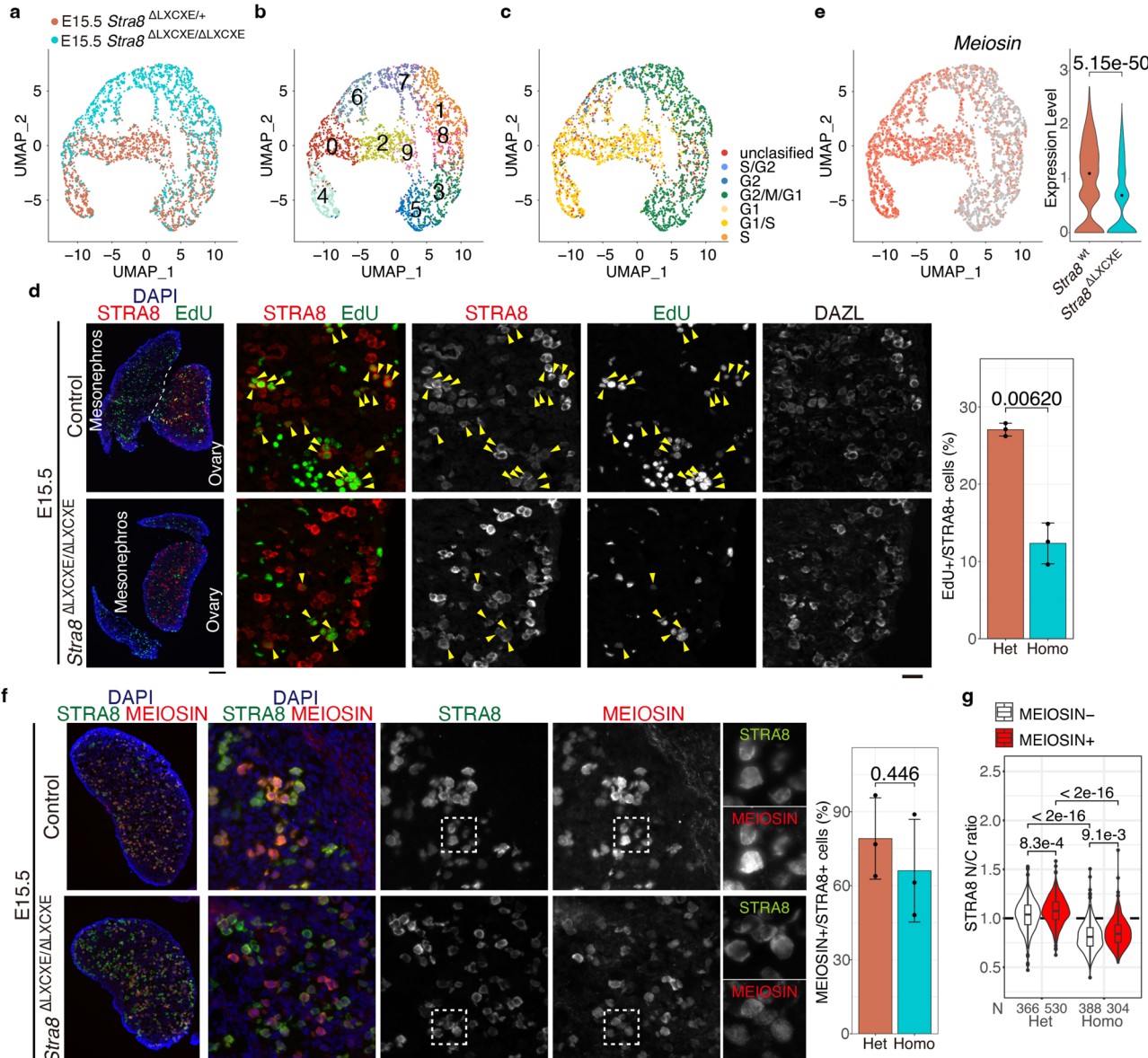

**Fig. 5 | Meiotic entry was delayed in the absence of STRA8-RB interaction.**
**a** Overlaid UMAP representation of scRNA-seq transcriptomes is shown for DDX4-expressing germ cells from heterozygous control and homozygous *Stra8* $^{\Delta LXCXE}$ $^{/\Delta LXCXE}$ $-3xFH-P2A-GFP$ knock-in ovaries at E15.5. **b** Clustering analysis of different gene expression patterns on UMAP-defined scRNA-seq transcriptomes. See also Supplementary Fig. 6 and Supplementary Data 4. **c** The estimated cell cycle status for STRA8-positive and STRA8-negative germ cells at E15.5 is shown on the UMAP plot. Individual cells are colored as indicated. **d** Sections from *Stra8* $^{\Delta LXCXE/\Delta LXCXE}$ and control ovaries at E15.5 were stained as indicated. Scale bars: 100 μm for whole sections (left); 20 μm (middle). EdU was injected for 2 h. DAZL represents a marker of germ cells. Quantification of EdU-incorporated cells per total number of STRA8-expressing germ cells in the control (*n* = 3) and *Stra8* $^{\Delta LXCXE/\Delta LXCXE}$ (*n* = 3) ovaries are shown on the right (bar plot of mean with +/− SD). *p* = 0.00620 (two-sided welch *t*-test). **e** Expression pattern and level of *Meiosin* are shown on the UMAP plot and

violin plot, respectively. *p* = 5.15 × 10$^{-50}$ (two-sided Wilcoxon rank-sum test). Black dots indicate median. **f** Sections from *Stra8* $^{\Delta LXCXE/\Delta LXCXE}$ and heterozygous control ovaries at E15.5 were stained as indicated. Scale bars: 100 μm for whole sections (left); 20 μm (middle). Enlarged images are shown on the right. Quantification of MEIOSIN-positive cells per STRA8-positive germ cells in the *Stra8* $^{\Delta LXCXE/+}$ (*n* = 3) and *Stra8* $^{\Delta LXCXE/\Delta LXCXE}$ (*n* = 3) ovaries are shown by bar plots of mean with SD (the most right). *p* = 0.446 (two-sided welch *t*-test). **g** Quantification of nuclear/cytoplasmic localization ratio of STRA8 in the MEIOSIN positive and negative cells at E15.5. Violin and Box plot of median and quantile. *p*-values were indicated in each plot (two-sided welch *t*-test with Bonferroni correction). Whiskers indicates maxima and minima. Black dots indicate outliers. N: number of cells examined. Three biologically independent female embryo derived cells were pooled and analyzed. See also Supplementary Figs. 6, Supplementary Data 3.

Thus, RB1 plays a role in suppressing mitotic cell proliferation in both male and female germ cell development.

In the canonical cell cycle, phosphorylation of RB1 by CDK leads to its dissociation from E2F and the activation of E2F-responsive genes[30]. Although the phosphorylated form of RB1 is decreased in male germ cells at E14.5[53], the extent to which CDK activity contributes to the transition from G1 to S phase in female germ cells is unknown. We found

that *Cdkn2a* which encodes alternative splicing products p16$^{INK4a}$ and p19$^{ARF}$[45,46] was upregulated in the pre-meiotic G1/S population of STRA8-expressing germ cells at E14.5 (Fig. 3g). Since p16$^{INK4a}$ and p19$^{ARF}$ directly or indirectly antagonize CDKs, CDK-mediated phosphorylation of RB1 may be less operative at the pre-mitotic G1/S transition in female germ cells, which may halt the mitotic cell cycle prior to meiotic initiation. Thus, it is possible that STRA8 may compensate for attenuated CDK

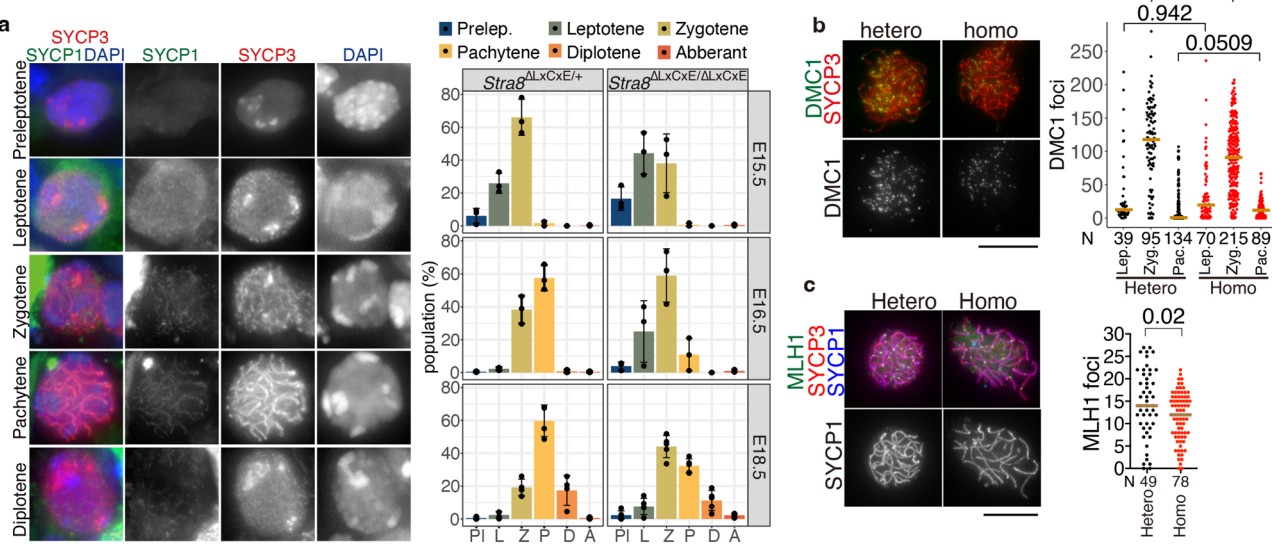

**Fig. 6 | Meiotic prophase was progressed with delay in *Stra8* ΔLXCXE/ΔLXCXE oocytes. a** Sections from *Stra8*ΔLXCXE/ΔLXCXE and control heterozygous ovaries at E15.5, E16.5 and E18.5 were stained as indicated. Immunostaining of SYCP1 (a marker of homolog synapsis) and SYCP3 (a marker of meiotic chromosome axis) showed the presence of germ cell population that are in the progress of meiotic prophase in *Stra8* ΔLXCXE/ΔLXCXE at E15.5, E16.5 and E18.5. Meiotic prophase population for *Stra8*ΔLXCXE/ΔLXCXE (E15.5: *n* = 2345 cells from 3 animals (264, 499 and 433 cells), E16.5: *n* = 1,557 cells from 3 animals (324, 807 and 426 cells), E18.5: *n* = 1642 cells from 5 animals (197, 330, 449, 317 and 349 cells)) and control heterozygous (E15.5: *n* = 1,335 cells from 3 animals (343, 678 and 314 cells), E16.5: *n* = 1,351 cells from 3 animals (459, 331 and 561 cells), E18.5: *n* = 2345 cells from 4 animals (350,198, 876 and 921 cells)) are shown in the graph (mean with SD). Scale bar: 5 μm. Het: *Stra8*ΔLXCXE/+,

Homo: *Stra8*ΔLXCXE/ΔLXCXE. **b** Chromosome spreads of *Stra8* ΔLXCXE/+ and *Stra8* ΔLXCXE/ΔLXCXE oocytes (pooled from E14.5, E16.5, E18.5 and PD0) were stained for DMC1, SYCP3, and SYCP1. Immunostained chromosome spread of zygotene spermatocytes are shown. The number of DMC1 foci is shown in the scatter plot with median (right). Statistical significance is shown by *p*-value (two-sided Wilcoxon rank-sum test). N: number of oocytes examined. Lep leptotene, Zyg. Zygotene, Pac. Pachytene. Scale bar: 10 μm. **c** Chromosome spreads of *Stra8* ΔLXCXE/+ and *Stra8* ΔLXCXE/ΔLXCXE oocytes (pooled from E18.5 and PD0) were stained for MLH1, SYCP3, SYCP1 and DAPI. The number of MLH1 foci is shown in the scatter plot with mean (right). Statistical significance is shown (two-sided Wilcoxon rank-sum test, *p* = 0.02). n: number of spermatocytes examined. Scale bar: 10 μm. N number of oocytes examined.

activity and enforce S phase progression by sequestering RB1 in female germ cells. Since *Stra8* ΔLXCXE/ΔLXCXE germ cells eventually passed through the pre-meiotic S phase with a delay at E15.5 (Fig. 5), STRA8-mediated sequestration of RB1 likely operates in parallel with CDK-mediated phosphorylation of RB1 in female germ cells. Nevertheless, STRA8–RB1 interaction is likely a dominant mechanism that promotes pre-meiotic S phase progression, which contrasts with continuous meiotic initiation in males (Supplementary Fig. 2).

The competency for meiotic initiation is regulated by BMP signaling synergistically with RA signaling in the fetal ovary[27]. In BMP signaling, ZGLP1 activates oogenic programs involving STRA8 expression prior to meiotic initiation in the fetal ovary[28]. In addition, epigenetic repressions, such as DNA methylation[54] and polycomb repressive complex (PRC)-mediated H3K27me3[42], imposed on meiotic genes are released prior to the initiation of meiosis in female germ cells. Our data suggest that STRA8 expression is required for the timely exit from the pluripotent state at E14.5 (Fig. 2) consistent with the previous observation that the upregulation of STRA8 expression in PRC1 KO female germ cells leads to precocious repression of pluripotent genes[42]. Therefore, STRA8 may directly or indirectly affect the repression of pluripotent status prior to meiosis entry in female mammals.

Co-expression of MEIOSIN and STRA8 is necessary for the decision of meiotic entry in germ cells, which is potentially regulated by RA signaling in pre-leptotene spermatocytes[9]. In males, *Meiosin* expression is inactive at the time of the first expression of STRA8 in spermatogonia but apparently upregulated at the second STRA8 expression upon exposure to RA[9]. Notably, our scRNA-seq analyses revealed that *Meiosin* was expressed later than *Stra8* in female germ cells. Moreover, *Meiosin* expression coincided with the pre-meiotic G1/S phase transition in the control STRA8-expressing oogonia, and was delayed in *Stra8* ΔLXCXE/ΔLXCXE oogonia at E14.5 (Fig. 4c and e). As *Stra8* ΔLXCXE/ΔLXCXE oogonia are capable

of responding to the endogenous RA signal at E14.5 (Fig. 2h), it seems that *Meiosin* expression in female germ cells is not directly regulated by RA. This finding is similar to a previous report that the RA response in *Meiosin* locus is seemingly slower than that of *Stra8* in GS cells[9]. Thus, *Meiosin* locus is activated later than *Stra8* expression and RA response in female germ cells. As the expression of *Meiosin* coincided with the G1/S transition and was delayed in the absence of STRA8–RB interaction (Figs. 4e, 5f), *Meiosin* expression may be dependent on the cell cycle state enforced by STRA8–RB interaction or the *Meiosin* locus may potentially be activated under E2F in female germ cells. However, the precise regulatory mechanisms of *Meiosin* expression in association with the cell cycle status of female germ cells require further investigation. Our findings indicate that multiple regulatory pathways synergistically coordinate germ cell development and meiotic initiation in fetal germ cells. Collectively, the present study sheds light on female-specific mechanisms of meiotic initiation in mammals.

## Methods
### Animals
LXCXE mutant *Stra8-3xFLAG-HA-p2A-GFP* knock-in (*Stra8* ΔLXCXE-*3FH* KI) and *Stra8-null GFP* knock-in (*Stra8* null GFP-KI) mice and other knockout and knock-in mice were congenic with the C57BL/6 background (age: embryonic 14-18 days old, postnatal day 0 and 17, 4- weeks and 8-weeks old). Wild type *Stra8-3xFLAG-HA-p2A-GFP* knock-in (*Stra8*wt-*3FH* KI) (age: embryonic 14-18 days old, postnatal day 0 and 17, 4- weeks and 8-weeks old) and *Stra8* KO (age: postnatal day 8, 4- weeks and 8-weeks old) mice, *Meiosin* KO mice (age: postnatal day 8) were generated as described in our previous study[9]. Male mice were used for immunoprecipitation of testis extracts (age: postnatal day 10-12 old), histological analysis of testes, and immunostaining of testes (age: postnatal day 8, 4- weeks and 8-weeks old). Female mice were used for Giemsa

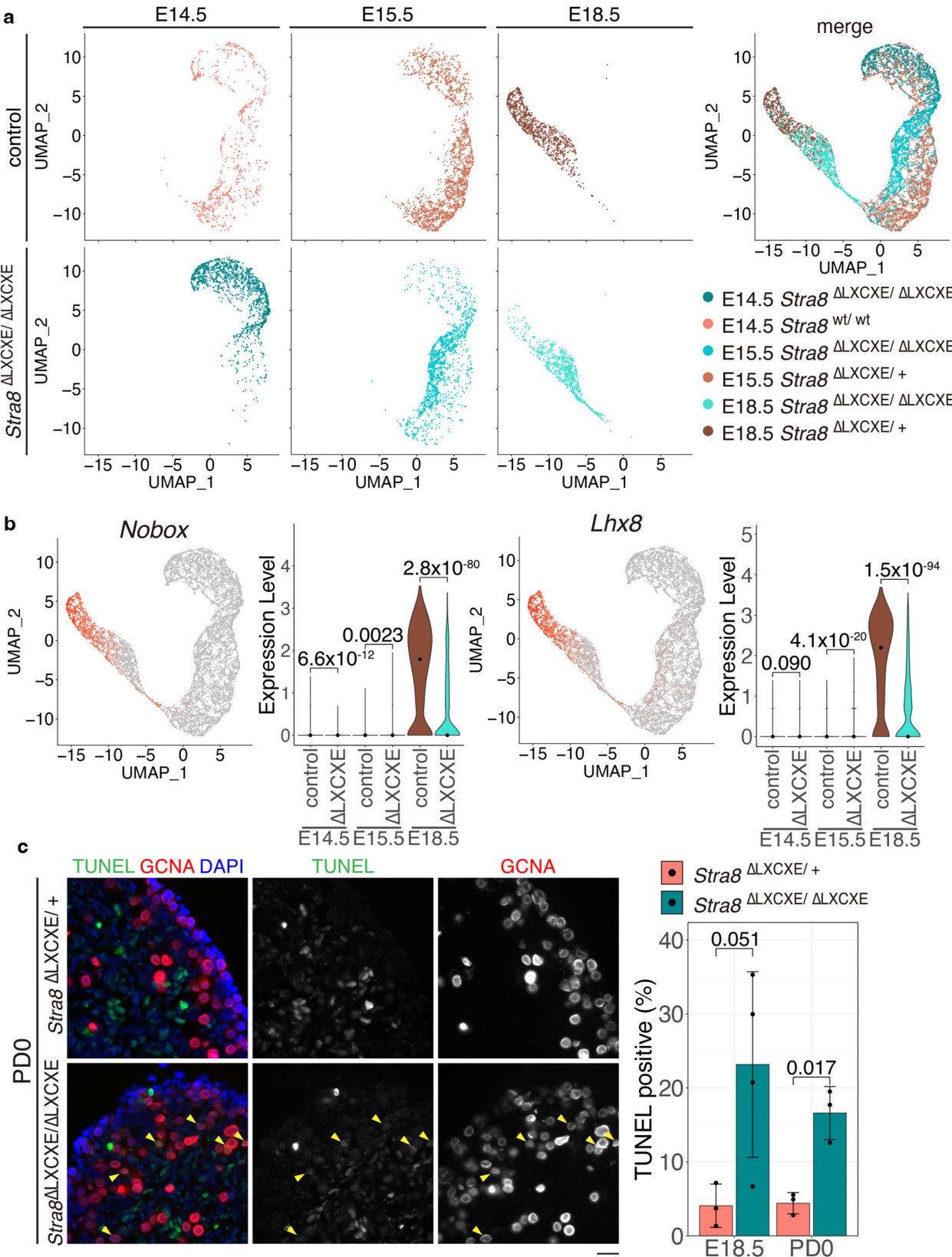

staining of oocyte chromosomes, histological analysis of the ovaries, immunostaining experiments, and sc-RNA-seq experiments (age: embryonic 14–18 days old, postnatal day 0 and 17, 4- weeks and 8-weeks old). Mvh/Ddx4-Venus transgenic mice: B6D2-Tg (Ddx4-Venus) 1Rbrc[55] were used for fluorescent sorting of female germ cells (age: embryonic 15, 18 days old). Whenever possible, each knockout animal was compared to littermates or age-matched non-littermates from the same colony, unless otherwise described. Housing conditions for the mice were under 12 h dark/12 h light cycle, ambient temperature at 20-23 degree C and humidity 40–60 %. Animal experiments were approved by the Institutional Animal Care and Use Committees of Kumamoto University (approval F28-078, A2022-001). Antibody

**Fig. 7 | Developmental progression was delayed in *Stra8* ^ΔLXCXE/ΔLXCXE^ oocytes.**
**a** UMAP representation of the combined analysis of scRNA-seq data of germ cells from control and homozygous *Stra8* ^ΔLXCXE/ΔLXCXE^-*3xFH-P2A-GFP* knock-in ovaries at E14.5, E15.5 and E18.5. Overlaid UMAP representation of scRNA-seq transcriptomes is shown on the right. **b** Expression patterns and levels are shown on the UMAP plot and violin plot, respectively, for the oocyte maturation/PPT genes *Nobox* and *Lxh8*. *p*-values were indicated in each plot. (two-sided Wilcoxon rank-sum test). Black dots indicate median. E14.5 control: E14.5 *Stra8* ^wt/wt^, E14.5 ΔLXCXE: *Stra8* ^ΔLXCXE/ΔLXCXE^, E15.5 control: *Stra8* ^ΔLXCXE /+^, E15.5 ΔLXCXE: *Stra8* ^ΔLXCXE/ΔLXCXE^, E18.5 control:

*Stra8* ^ΔLXCXE/+^, E18.5 ΔLXCXE: *Stra8* ^ΔLXCXE/ΔLXCXE^. **c** Sections from *Stra8* ^ΔLXCXE/ΔLXCXE^ and heterozygous control ovaries at E18.5 and PD0 were subjected to TUNEL assay with immunostaining for GCNA. Quantification of TUNEL-positive cells (yellow arrowed) per total number of GCNA-positive germ cells in the control (*n* = 3 for E18.5, *n* = 3 for PD0) and *Stra8*^ΔLXCXE/ΔLXCXE^ (*n* = 4 for E18.5, *n* = 3 for PD0) ovaries is shown on the right (barplot of mean with +/− SD). *p* = 0.051 for E18.5 and *p* = 0.017 for PD0 (two-sided Welch *t*-test). Scale bar: 20 µm. See also Supplementary Fig. 7, Supplementary Data 5.

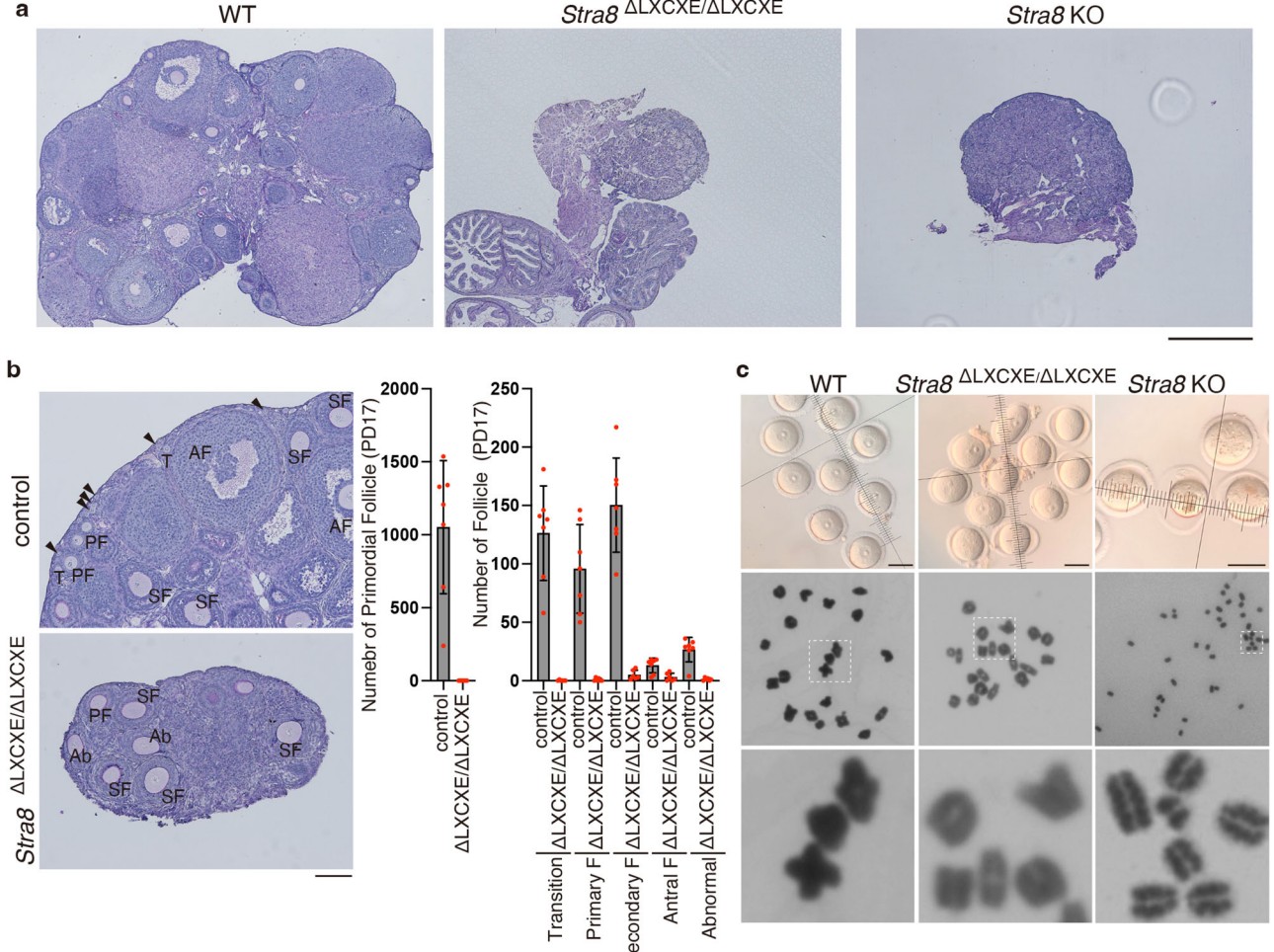

**Fig. 8 | Premature loss of oocyte pool in *Stra8*-mutant ovaries with RB-binding deficiency. a** Hematoxylin and eosin staining of the sections from wild type (WT), LXCXE mutant *Stra8* ^ΔLXCXE/ΔLXCXE^-*3FH* KI homozygous and *Stra8* KO ovaries (eight-week-old). Scale bar: 500 µm. **b** Hematoxylin and eosin staining of the sections from *Stra8* LXCXE mutant homozygous (*Stra8* ^ΔLXCXE/ΔLXCXE^) and the control (*Stra8*^ΔLXCXE/+^) ovaries at postnatal day 17 (left). Numbers of primordial follicles (shown by arrowhead) in *Stra8* LXCXE mutant homozygous (6 ovaries pooled from *n* = 5 animals) and control heterozygous (7ovaries pooled from *n* = 5 animals) ovaries are shown in the middle (bar graph of mean with +/− SD). Numbers of

transition (T), primary (PF), secondary (SF), antral (AF) and abnormal (Ab) follicles in *Stra8* LXCXE mutant homozygous and control heterozygous ovary are shown on the right (bar graph with SD). Scale bar: 100 µm. **c** GV oocytes isolated from WT, *Stra8*^ΔLXCXE/ΔLXCXE^ females and oocyte-like cells of *Stra8* KO female (upper). Scale bars: 50 µm. Giemsa staining of metaphase I bivalents in WT, *Stra8* LXCXE mutant homozygous oocytes (x100 magnification), and metaphase-like univalent in *Stra8* KO oocyte-like cells (x63 magnification) (middle). Enlarged images are shown on the bottom.

production using rabbit and guinea pig was done by a contractor (Kiwa Laboratory Animals Co., Ltd.). The handling of these animals was carried out according the contractor's animal experimental protocols.

### Generation of LXCXE mutant *Stra8-3xFLAG-HA-p2A-GFP* knock-in mouse and genotyping

LXCXE mutant *Stra8-3xFLAG-HA-p2A-GFP* knock-in (*Stra8* ^ΔLXCXE^-*3FH* KI) mouse was generated by introducing Cas9 protein (317-08441; NIPPON GENE, Toyama, Japan), tracrRNA (GE-002; FASMAC, Kanagawa, Japan),

synthetic crRNA (FASMAC) and ssODN into *Stra8-3xFLAG-HA-p2A-GFP* knock-in[9] fertilized eggs using electroporation. The synthetic crRNA was designed to direct ACAGATCGTCAAAGGTCTCC (agg) of the *Stra8* Exon 9. ssODN: 5'- ACCAGAGCCCCCAGATGATGATGATGCAATGTT GgcGAAGgcgCTGGcGACCTTTGACGATCTGtccGGAGACTACAAAGAC CATGACGG-3' was used as a homologous recombination template.

The electroporation solutions contained 10 µM of tracrRNA, 10 µM of synthetic crRNA, 0.1 µg/µl of Cas9 protein, ssODN (1 µg/µl) in Opti-MEM I Reduced Serum Medium (31985062; Thermo Fisher Scientific).

Electroporation was carried out using the Super Electroporator NEPA 21 (NEPA GENE, Chiba, Japan) on Glass Microslides with round wire electrodes, 1.0 mm gap (45-0104; BTX, Holliston, MA). Four steps of square pulses were applied (1, three times of 3 mS poring pulses with 97 mS intervals at 30 V; 2, three times of 3 mS polarity-changed poring pulses with 97 mS intervals at 30 V; 3, five times of 50 mS transfer pulses with 50 mS intervals at 4 V with 40% decay of voltage per each pulse; 4, five times of 50 mS polarity-changed transfer pulses with 50 mS intervals at 4 V with 40% decay of voltage per each pulse).

The *Stra8* ΔLXCXE-*3FH* KI allele was identified by PCR using the following primers; St8-24996F: 5′-AGGCCCAGCATATGTCTAACATC AG-3′ and

St8-29564R: 5′- AGAAGGCTTTTGGAAGCAGCCTTTC-3′

for the knock-in allele (4586 bp) and the wild-type allele (3668 bp). The PCR amplicons were verified by sequencing. For avoiding potential mosaicism of the mutant allele, the F0 founders were backcrossed with C57BL/6 to segregate *Stra8* ΔLXCXE-*3FH* KI allele and establish heterozygous lines. Since homozygous female mouse with *Stra8* ΔLXCXE-*3FH* KI allele was infertile, female mice were maintained as *Stra8* ΔLXCXE-*3FH* KI heterozygous mutant and male mice were maintained as *Stra8* ΔLXCXE-*3FH* KI homozygous or heterozygous mutant. Primers are listed in Supplementary Data 6.

### Generation of *Stra8*-null-*GFP* knock-in mouse and genotyping

*Stra8*-null *GFP* knock-in (*Stra8* null GFP-KI) allele was generated by removing all the *Stra8*-coding exons (Exon2-Exon9) from *Stra8* wt-*3xFLAG-HA-p2A-GFP* knock-in allele. Cas9 protein, tracrRNA, synthetic crRNA (FASMAC) and ssODN were introduced into *Stra8-3xFLAG-HA-p2A-GFP* knock-in fertilized eggs using electroporation. The synthetic crRNAs were designed to direct TAGATTATAATGGCCACCCC(TGG) of the *Stra8* Exon 2, and CCCGGGCCTATGGTGAGCAA(GGG) of the border at *p2A-GFP*. ssODN: CGACCAGGATGGGCACCACCCCGGTGAACAGCTC CTCGCCCTTGCTCACGGCCATTATAATCTACAACATAATATACCCCC TTTAGCCTTGCATAATA was used as a homologous recombination template.

The *Stra8* null GFP-KI allele was identified by PCR using the following primers; St8-delHLH 1 F: 5′-CAAGGCTAAAGGGGGGTATATTATG-3′ and

St8-29564R: 5′- AGAAGGCTTTTGGAAGCAGCCTTTC-3′

for the knock-in allele (905 bp). The PCR amplicons were verified by sequencing. For avoiding potential mosaicism of the mutant allele, the F0 mutant founders were backcrossed with C57BL/6 to segregate *Stra8* null GFP-KI allele and establish heterozygous lines. Since homozygous mouse with *Stra8* null GFP-KI allele was infertile, both male and female mice were maintained as *Stra8* null GFP-KI heterozygous mutant. Primers are listed in Supplementary Data 6.

### EdU labeling of mouse ovaries

Pregnant female mice were injected intraperitoneally with 200 μl of 2 mg/ml EdU (Wako). Embryonic ovaries were collected 2 h after the injection. EdU-incorporated cells on the ovary sections were envisioned using a Click-iT EdU imaging kit (Thermo-Fisher).

### Preparation of testis extracts

Testis chromatin-bound and -unbound extracts were prepared as described previously[9,56]. Briefly, testicular cells were suspended in low salt extraction buffer (20 mM Tris-HCl pH 7.5, 100 mM KCl, 0.4 mM EDTA, 0.1% TritonX100, 10% glycerol, 1 mM β-mercaptoethanol) supplemented with Complete Protease Inhibitor (Roche). After homogenization, the soluble chromatin-unbound fraction was separated after centrifugation at 100,000 g for 30 min. The chromatin bound fraction was extracted from the insoluble pellet by high salt extraction buffer (20 mM HEPES-KOH pH 7.0, 400 mM KCl, 5 mM $MgCl_2$, 0.1% Tween20, 10% glycerol, 1 mM β-mercaptoethanol) supplemented with Complete Protease Inhibitor. The solubilized chromatin fraction was collected after centrifugation at 100,000 g for 30 min at 4 °C.

### Immuno-affinity purification

The 3xFLAG-HA tagged version of WT STRA8 and the mutant STRA8ΔLXCXE proteins were immunoprecipitated from testis extracts of *Stra8* wt-*3FH* KI and *Stra8* ΔLXCXE-*3FH* KI mice by tandem IPs with FLAG and HA antibodies as described previously[9,56]. For the negative control, testis extracts from WT male that do not possess 3xFLAG-HA tagged version of STRA8 (non-tagged control) were used. Immuno-affinity purification was performed with anti-FLAG M2 monoclonal antibody-coupled magnetic beads (Sigma-Aldrich) from the testis chromatin-bound fraction of *Stra8* ΔLXCXE-*3FH* KI and *Stra8*wt-*3FH* KI mice (10 to 12-day old). For negative control, mock immuno-affinity purification was done from the testis chromatin-bound fractions from the age-matched wild type mice. The beads were washed with high salt extraction buffer for chromatin-bound proteins and low salt extraction buffer for chromatin-unbound proteins. The anti-FLAG-bound proteins were eluted by 3xFLAG peptide (Sigma-Aldrich). The second immuno-affinity purification was performed with M132-11 anti-HA-tag mAb-Magnetic Beads (MBL). The bead-bound proteins were eluted with 40 μl of elution buffer (100 mM Glycine-HCl pH 2.5, 150 mM NaCl), and then neutralized with 4 μl of 1 M Tris-HCl pH 8.0.

For immunoprecipitation of endogenous RB1 from *Stra8*wt-*3FH* KI FLAG-IP, 5 μg of affinity-purified rabbit anti-RB1-N and control rabbit IgG antibodies were crosslinked to 50 μl of protein A-Dynabeads (Thermo-Fisher) by DMP (Sigma). The antibody-crosslinked beads were added to the FLAG-IP prepared from *Stra8*wt-*3FH* KI testes (10 to 11-day-old). The beads were washed as described above. The bead-bound proteins were eluted as described above. The immuno-precipitated proteins were run on 4–12% NuPAGE (Thermo-Fisher) in MOPS-SDS buffer and immunoblotted. Immunoblot image was developed using ECL prime (GE healthcare) and captured by FUSION Solo (VILBER).

### Antibodies

The following antibodies were used for immunoblot (IB) and immunofluorescence (IF) studies: rabbit anti-PLZF (IF, 1:1000, Abcam: ab189849), rabbit anti-DAZL (IF, 1:1000, ab34139), rabbit anti-HA (IB, 1:1000, Abcam: ab9110), mouse anti-HA 12CA5 monoclonal Antibody (IF, 1:100, Roche: AB_514505), rabbit anti-H3S10P (IF, 1:2000, Abcam: ab5176), rabbit anti-SYCP1 (IF, 1:1000, Abcam ab15090), rabbit anti-DMC1 (IF, 1:500, Santa Cruz: SC-22768), mouse anti-MLH1 (IF, 1:500, BD Biosciences: 551092), rabbit anti-Actin (IB, 1:1000, Sigma A2066), rabbit anti-GFP (IF, 1:1000, ab6556), rat anti-TRA98 (IF, 1:1000, ab82527), rabbit anti-p107 (IB, 1:1000, Santa Cruz: SC-318), rabbit anti-FOXO3 (IF, 1:200, CST 2497), rat anti-NANOG (IF, 1:1000 Thermo: eBioMLC-51), rat anti-SYCP3 and guinea pig anti-SYCP3, rabbit and rat anti-STRA8, rabbit and gunia pig anti-MEIOSIN N-terminal (a.a. 1-224) and rabbit and gunia pig anti-MEIOSIN C-terminal (a.a. 405-589) as described previously[9].

### Antibody production

Polyclonal antibodies against mouse STRA8 (a.a. 1-393) were generated by immunizing gunia pig. Polyclonal antibodies against mouse RB1-C terminal (a.a. 479-921) were generated by immunizing rabbit. All His-tagged recombinant proteins were produced by inserting cDNA fragments in-frame with pET19b (Novagen) in *E. coli* strain BL21-CodonPlus(DE3)-RIPL (Agilent), solubilized in a denaturing buffer (6 M HCl-Guanidine, 20 mM Tris-HCl pH 7.5) and purified by Ni-NTA (QIAGEN) under denaturing conditions. The antibodies were affinity-purified from the immunized serum with immobilized antigen peptides on CNBr-activated Sepharose (GE healthcare).

### Histological analysis

For hematoxylin and eosin staining, testes, epididymis and ovaries were fixed in Bouin solution, and embedded in paraffin. Sections were prepared on MAS-GP typeA-coated slides (Matsunami) at 6 μm

thickness. The slides were deparaffinized and stained with hematoxylin and eosin.

## Immunofluorescence microscopy of testis and ovary

For immunofluorescence staining, the testes were embedded in Tissue-Tek O.C.T. compound (Sakura Finetek) and frozen at −80 °C. The embryonic ovaries were fixed in 4% PFA for 30 min at 4 °C and submerged sequentially in 10%, 20% and 30% sucrose at 4 °C, then the fixed ovaries were embedded in Tissue-Tek O.C.T. compound and frozen at −80°C. Cryosections were prepared on the MAS-GP typeA-coated slides (Matsunami) at 6 µm thickness, and then air-dried. The testes were fixed in 4% paraformaldehyde in PBS at pH 7.4. The sections were blocked for 30 min in 3% BSA containing 0.1% TritonX100 in TBS (TBS-T), and incubated at 4 °C with the primary antibodies in a blocking solution. After three washes in TBS-T, the sections were incubated for 90 min at room temperature with Alexa-dye-conjugated secondary antibodies (1:1000; Invitrogen) in TBS-T. DNA was counterstained with Vectashield mounting medium containing DAPI (Vector Laboratory).

## Imaging

Immunostaining images were captured with DeltaVision (GE Healthcare). The projection of the images was processed with the SoftWorx software program (GE Healthcare). All images shown were Z-stacked. For counting seminiferous tubules and embryonic ovaries, immunostaining images were captured with BIOREVO BZ-X710 (KEYENCE), and processed with BZ-H3A program. XY-stitching capture by 40x objective lens was performed for multiple-point color images using DeltaVision. Images were merged over the field using SoftWorx software program. Bright field images were captured with OLYMPUS BX53 fluorescence microscope and processed with CellSens standard program.

## In vitro oocyte culture and Giemsa staining of metaphase chromosome spread

Ovaries collected from 4-week-old female mice were used after 46 to 48 h of treatment with 5 IU of pregnant mare serum gonadotropin. GV oocytes from WT and $Stra8^{\Delta LXCXE/\Delta LXCXE}$-3FH KI, and oocyte-like cells from Stra8 KO were isolated by puncturing the follicles in M2 medium (Sigma). The GV oocytes and GV oocyte-like cells were cultured in M16 medium (Sigma) in a 5% $CO_2$ atmosphere at 37 °C for 5–6 h. For Giemsa staining of metaphase chromosome spread, oocytes and oocyte-like cells were exposed to 0.5% Pronase (Sigma) to remove the zona pellucida, and treated in hypotonic buffer containing 1% sodium citrate/0.1% PVA for 15 min. The oocytes and oocyte-like cells were placed on the slides, fixed in the Carnoy's Fixative (75 % Methanol, 25% Acetic Acid) and stained in 3% Giemsa solution for 30 min.

## WIN 18,446 and retinoic acid treatment

Neonatal male mice were injected subcutaneously with 100 µl of 10 mg/ml WIN 18,446 (5% DMSO/95% corn oil, Cayman Chemical) daily during 2dpp-6dpp to block spermatogonial differentiation. At 7dpp, mice receiving consecutive WIN 18,446 treatments were injected intraperitoneally with 100 µl of 2.5 mg/ml all-trans Retinoic Acid (RA) (10% DMSO/90% corn oil), followed by testes collection at 8dpp.

## Single-cell RNA-sequencing

Embryos were collected at E14.5 (from $Stra8^{wt/wt}$-3FH KI, $Stra8^{\Delta LXCXE/\Delta LXCXE}$-3FH KI and $Stra8^{nullGFP/nullGFP}$ mice), and at E15.5 and at E18.5 (from $Stra8^{\Delta LXCXE/+}$-3FH KI/Ddx4-Vienus Tg, $Stra8^{\Delta LXCXE/\Delta LXCXE}$-3FH KI/Ddx4-Vienus Tg) after IVF. After genotyping, single-cell suspensions were prepared by incubating ovaries with Accutase (Innovative Cell Technologies, Inc.) for 5 min at 37 °C. Single-cell suspensions were pooled from embryonic ovaries. After incubation, DMEM with 10% FBS was added to block Accutase and aggregate was disrupted by pipetting. Then, cell suspensions were filtered through a 35-µm cell strainer sieve (BD Bioscience). Cells were collected by centrifugation and re-suspended in PBS containing 0.1% BSA, and GFP positive cells were collected using Cell Sorter SH800 (SONY). Collected cells were re-suspended in DMEM containing 10% FBS. Resulting approximate 1000 - 2000 single-cell suspensions were loaded on Chromium Controller (10X Genomics Inc.). Single cell RNA-seq libraries were generated using Chromium Single Cell 3' Reagent Kits v3 following manufacturer's instructions, and sequenced on an Illumina HiSeq X to acquire paired end 150 nt reads. The number of used embryos, the total numbers of single cells captured from ovaries, mean depth of reads per cell, average sequencing saturation (%), the number of detected genes, median UMI counts/cell, total number of cells before and after QC, are shown in Figure S4. Sequencing data are available at DDBJ Sequence Read Archive (DRA) under the accession DRA013182 and DRA015395.

## Statistical analysis of scRNA-seq

Fastq files were processed and aligned to the mouse mm10 transcriptome (GENCODE vM23/Ensembl 98) using the 10X Genomics Cell Ranger v 4.0 pipeline. Further analyses were conducted on R (ver.3.6.2) (R Core Team, 2019) via RStudio (ver.1.2.1335) (RStudio Team, 2018). Quality assessment of scRNA-seq data and primary analyses were conducted using the Seurat package for R (v3.2.2)[57,58]. Contaminated somatic cells were excluded, and only the cells that expressed more than 200 genes were used for further analysis to remove the effect of low-quality cells. The scRNA-seq data were merged and normalized using SCTransform function built in Seurat. The dimensional reduction analysis and visualization of cluster were conducted using RunUMAP function built in Seurat. The clustering of cells was conducted using FindNeighbors and FindClusters built in Seurat with default setting. Determination of differentially expressed genes (DEGs) was performed using FindAllMarkers function built in Seurat with following options, only.pos = T, min.pict = 0.25 and logfc.threshold = 0.25, for the identification of marker genes in each cluster, and FindMarker function built in Seurat with default settings to characterize the arbitrary group pf cells. RNA velocity analysis was conducted using the RNA velocyto package for python (Python 3.7.3) and R with default settings[41], and visualized on UMAP plots built in Seurat.

## Gene enrichment analysis

Gene enrichment analyses were performed using Metascape[59] with default settings and the results were visualized using R. To characterize the feature of each cluster, top100 representative genes in each cluster were used.

## Cell cycle scoring analysis

Cell cycle estimation using scRNA-seq data was conducted as previously reported[60,61] with some modification that is suitable for germ cell analysis[62]. Cell cycle scoring analysis of scRNA-seq data approximately predicts cell cycle status according to fluctuation of cell cycle-associated genes. The reference gene set is pre-defined according to mRNA expression levels that correlate with fractions of cell cycle stages (G1/S, S, G2, G2/M, M/G1) (Supplementary Data 3)[63]. Cell cycle status is estimated to a given single cell according to fluctuation of cell cycle-related reference genes in the scRNA-seq data and shown in the heat map (Fig. 4c). Among the pre-defined reference gene set in the scRNA-seq data, those genes weakly correlated with any of cell cycle phase were excluded (R < 0.2). The average expression of the reference genes was calculated to define the score for cell cycle phase in each single cell. The calculated cell cycle scores were scaled to acquire a pattern of cell cycle phase-specific scores. Then, each pattern of phase-specific scores was compared across all patterns, those single cells with the most similar pattern was clustered. Those single cells, that are

assigned as G1, G1/S and S, can be separately recognized by distinct gene expression patterns. In contrast, most of those single cells, that once progress into G2 phase, are difficult to define a particular cell cycle phase due to persistent gene expression pattern across G2, M, G1 phases. Therefore, those single cells, which cannot be assigned to any particular stage of G2, M or G1 phase, were categorized as G2/M/G1. Those single cells that cannot be assigned to any particular cell cycle stage were categorized as unclassified.

## Quantification of nuclear/cytoplasmic localization of STRA8

Quantification of STRA8 localization was conducted using ImageJ software. The mask for STRA8 expressing cells was generated using Auto local threshold function built in ImageJ with Bernsen method. To optimize the whole STRA8 signal mask, Erode, Dilate and Fill Holes functions were used, if necessary. DAPI binary image was generated using Auto local threshold function with Mean method to define the nuclear mask. To generate the nuclear STRA8 signal mask, whole STRA8 signal mask was overlayed to the nuclei of STRA8 expressing cells. Nuclear STRA8 signal mask was subtracted from whole STRA8 signal mask to generate cytoplasmic STRA8 signal mask. Using the nuclear and cytoplasmic STRA8 signal mask, the mean intensity of STRA8 signals per pixel of nuclear and cytoplasmic regions were measured. The nuclear/cytoplasmic ratios of STRA8 signals were calculated and plotted using R.

## Reporting summary

Further information on research design is available in the Nature Portfolio Reporting Summary linked to this article.

## Data availability

Mouse lines generated in this study have been deposited to Center for Animal Resources and Development (CARD ID2610 for STRA8 LXCXE mutant *Stra8* $^{\Delta LXCXE}$ *-3xFLAG-HA-p2A-GFP* knock-in mouse, CARD ID3091 for *Stra8-null GFP* knock-in (*Stra8* $^{null\,GFP}$-KI) mouse). The antibodies are available upon request. There are restrictions to the availability of antibodies due to the lack of an external centralized repository for its distribution and our need to maintain the stock. We are glad to share antibodies with reasonable compensation by requestor for its processing and shipping. All unique/stable reagents generated in this study are available from Kei-ichiro Ishiguro with a completed Materials Transfer Agreement. Further information and requests for resources and reagents should be directed to and will be fulfilled by Kei-ichiro Ishiguro (ishiguro@kumamoto-u.ac.jp). All data supporting the conclusions are present in the paper and the supplementary materials. A reporting summary for this article is available as Supplementary Information file. The source data (for Fig. 2d, Fig. 2e, Fig. 2g, Fig. 3d, Fig. 3e, Fig. 3f, Fig. 4c, Fig. 4d, Fig. 4e, Fig. 5d, Fig. 5e, Fig. 5f, Fig. 5g, Fig. 6a, Fig. 6b, Fig. 6c, Fig. 7b, Fig. 7c, Fig. 8b, Supplementary Fig. 2c, Supplementary Fig. 3b, Supplementary Fig. 5b, Supplementary Fig. 6a, Supplementary Fig. 7d) are provided with this paper.

Raw sequence data generated in this study were publicly available as of the date of publication. Sequencing data have been deposited in DDBJ Sequence Read Archive (DRA) under the accession DRA013182 for scRNA-seq data of E14.5 germ cells and DRA015395 for the scRNA-seq data of E15.5 and E18.5 germ cells. Source data are provided with this paper. The original images in the figures have been deposited in Figshare https://doi.org/10.6084/m9.figshare.24205230.

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

## Acknowledgements

The authors thank Kazumasa Takemoto, Kumi Matsuura, Akiko Miyoshi, and Sakie Iisaka for technical assistance. This work was supported in part by KAKENHI grant (#20K22638) to R.S.; KAKENHI grants (#19H05743, #20H03265, #20K21504, #22K19315, #23H00379, AdAMS # 16H06276, AdAMS #22H04922) to K.I.; Grant from AMED PRIME (23gm6310021h0003) to K.I.; Grants from The Mitsubishi Foundation; The Sumitomo Foundation; The Naito Foundation; Astellas Foundation for Research on Metabolic Disorders; Takeda Science Foundation to K.I.; and the program of the Research for Inter-University Research Network for High Depth Omics, IMEG, Kumamoto University to K.I.

## Author contributions

R.S. performed the scRNA-seq and embryonic gonadal experiments and wrote the draft of the manuscript. Y.K. performed the ovarian follicle analysis. N.T. and K.A. generated the mutant mice and performed IVF. N.T. and K.I. performed the GV oocyte experiments. S.F. performed the histological analysis. K.Y. and S.U. assisted with the scRNA-seq. H.N. assisted in the construction of the expression vectors and ES cell experiments. K.I. conducted the study, performed the experiments in mice, and wrote the manuscript. The experimental design and data interpretation were conducted by R.S. and K.I.

## Competing interests

The authors declare no competing interests.
