## [Peer Review File · Nature Communications]

STRA8–RB interaction is required for timely entry of meiosis in mouse female germ cellsReviewer #1 (Remarks to the Author):

Key results

This team has previously revealed the importance of MEIOSIN, which interacts with the critical pre-meiotic transcription factor STRA8 and is necessary for meiotic onset and progression (Ishiguro et al 2020). Here they show that STRA8 also binds to RB1 and related protein p107, and that this binding is independent of the STRA8/MEIOSIN interaction and uses a different domain of STRA8. They use in vivo mutation of the evolutionarily-conserved RB1 binding site in STRA8 (referred to as LXCXE; X is replaced by Alanine) to reveal that STRA8/RB interaction ensures timely meiotic onset, in terms of both meiotic S phase entry and meiotic gene activation. They also show that mutation of the RB binding site leads to precocious depletion of the oocyte pool. The phenotype of the LXCXE mutant is, however, not as severe as the Stra8 KO (the latter fails to enter meiosis at all, whereas the former is delayed in meiotic progression). The model suggests that STRA8 sequesters RB from E2F at the G1/S transition, thereby ensuring pre-meiotic G1/S phase transition. It seems that nuclear localization of STRA8 depends on both RB1 (shown here) and MEIOSIN (shown previously, Ishiguro et al 2020). The significance is potentially limited by the fact that meiotic onset and progression is only slowed in the absence of STRA8-RB1 complex formation.

Overall assessment

The quality of the data seems good. A great deal of work has been done and, technically, I have no reason to doubt that it is sound. The greatest problem with the manuscript is that it is very loosely written and hard to follow.

Major points to address

1. Do we know that STRA8 co-immunoprecipitates with RB and p107 in the fetal ovary?
2. Why would some STRA8 bind MEIOSIN and some STRA8 bind RB? What would prevent all three being present in one complex?
3. Why would MEIOSIN expression be weaker in the Stra8DLxCxE-3FHKI than in the Stra8 wt -3FH KI control (Figure 1D)? Couldn't this be the reason there is more cytoplasmic STRA8, rather than a role of RB1 here? (lines 139 to 144)
4. Why do you think Zglp1 expression differs between the wildtype and the LxCxE mutant (Figure 2G)? This does not seem consistent with the idea that ZGLP1 is downstream of BMP signalling but not downstream of RA or STRA8.
5. Oct4 not included in Figure 2D and is not mentioned as a gene related to stem cell maintenance (line 170) - does it act differently to Nanog, Sox2, Esrrb and Klf5?
6. The story line becomes a little muddled when the Stra8 KO is included. I think it would be better to focus on the LxCxE mutant.
7. Line 189 - you say that STRA8 is required for the germ cells to become 'competent for meiosis' but I do not see how this terminology is justified. In fact, the actions of BMP signalling might be more about 'competence' - in the absence of STRA8 the BMP targets are induced, but yet there is no meiosis as there is no STRA8.
8. Meiosin expression depends on the cell cycle status of G1/S transition in female germ cells (line 252) (what does this mean? that this is downstream of STRA8? this does not agree with earlier paper where it was claimed that both are induced by RA but in independent pathways?). Also MEIOSIN protein is lower in the LxCxE mutant. What does this mean?
9. If STRA8 sequesters RB1 away from E2F to 'depress the expression of E2F-responsive genes' then why is the expression of this gene set lower in the LXCXE mutant? (Figure 3F)?

Minor points to address

1. The first sentence of the abstract is a little obtuse - 'by sexually different mechanisms in mice'. This is also stated line 45. Suggests that the process of meiosis is different in different sexes, rather than the timing of meiosis onset.
2. LXCXE is often written as LxCxE. It would be good if one version could be used throughout, including in the figures. Why not LACAE?
3. In general, the referencing is sloppy. References presented in Introduction line 40 seem quite arbitrary (eg Oulad-Abdelghani 1996 and Menke 2003 did not link STRA8 and meiotic entry). Later at line 53/54, Saitou and Yamaji 2012 and Yokobayashi et al 2013 reference 'downregulation of pluripotency genes' at the time of meiotic initiation (but the first of these does not mention this

and the second only in passing). Line 57 - not sure that Cordeiro et al 2015 is appropriate here (this ref about follicle activation not meiotic onset) - there are a number of other references where the anterior-posterior wave were revealed. There are many other examples - it is important to be precise in the literature. Reference the primary publications where possible and reviews where necessary, preferably noting that it is a review.

4. 'ALDH1a families' is not clear

5. Line 68 - 'how initiation of meiosis is coordinated with cell cycle so that it coincides with pre-meiotic S phase' - wouldn't it be inducing pre-meiotic S phase rather than just coinciding?

6. Line 75 - 'Here we present sexually different mechanisms of meiotic initiation in mice' doesn't seem to make sense, nor to overview the findings of the study.

7. Figure 1B has both Stra8DRb-3FHKI as well as Stra8DLxCxE-3FHKI shown (I think these are the same thing, and the second more consistent with the manuscript).

8. Throughout, Rb should be referred to by its correct name, Rb1

9. Line 116 - The expression of Stra8 in the testis was first shown by Oulad-Abdelghani 1996.

10. Line 133 - I do not think you can say that 'FLAG-HA-tagged STRA8 was expressed at similar timing from both Stra8 wt -3FH KI and Stra8 Δ LxCxE-3FH KI alleles' when you only looked at a solitary timepoint, E14.5.

11. Line 167 - I am not sure that you can say 'distinct clusters'. This would be true of the wildtype vs KO, but the wildtype vs LxCxE homs seem to reveal different clustering but not 'distinct clusters'.

12. Figure 2B - says verocity not velocity

13. I am not sure why it would be 'remarkable' that pluripotency-associated gene expression would remain high in the Stra8 KO cells? I think this is similar to observations by many other studies (eg see the reciprocal expression patterns in Bullejos and Koopman 2004 Mol Repro and Dev).

14. Wu et al 2016 does not belong here - this was regarding SMAD4 deletion not BMP action.

15. Line 195 - says 11 clusters, but only ten are shown in Figure S5.

16. Figure 3G - why is quantitation not shown for this particular gene as it is for the others?

17. Line 307 - the word 'between' here is not appropriate (suggests communication?). 'Gene expression changes as primordial follicles transition to primary follicles'?

18. Line 26, 309, 328, 364, 375 - not sure what you mean by 'unscheduled' - sounds like ectopic or early but I think you mean abnormal or delayed.

19. Line 310 - 'Oocytes accompanied' should possibly be 'oocytes displayed'

20. Line 328 - why would we consider this an 'indirect consequence'? Surely if delayed at E14.5 this might lead to delay at E18.5?

21. Line 341 - this implies that the Stra8 KO is similar to the Stra8 LxCxE mutant in terms of fertility phenotype - but I think the follicles are already absent at birth for the former?

22. Line 343 - 'starved'? lost?

23. Line 396 - additional and original reference is Bowles et al, 2006

24. Line 397 - I do not understand what you mean by 'this could presumably be due to actively driving cell cycle in the passage from spermatogonia to spermatocyte, which is in contrast to the oogonia'.

25. Line 419 - you seem to have left out the step where p16/p19 antagonise CDKs

26. Line 421 - 'the overall expression levels of E2F-target genes were relatively lower in pre-meiotic G1/S (Cluster 5) compared to mitotic G1/S (Cluster 6) - Fig 3A, E.' I am very confused - Figure 3 doesn't show clusters at all. And overall expression of E2F target genes is lower in the LXCXE mutant in Figure 3F.

27. Expression level of the Cdkn2a was reciprocally higher in pre-meiotic G1/S than in mitotic G1/S (Fig. 3F) - ??? this figure compares Stra8 Wt with the LXCXE mutant??

28. There is no Figure 4F shown.

Reviewer #2 (Remarks to the Author):

This is a detailed analysis of factors involved in initiation and progression of meiosis in female mice. Combining transcriptomic analyses with different STRA8 mutations (most importantly, one affecting STRA8-RB binding), the authors provide an exhaustive analysis of the regulation of

meiotic initiation in mammalian females and its progression through the fetal stages of meiosis. This adds important new insights to the way in which oocytes are formed, and the authors are to be congratulated on characterizing the complicated set of interactions underlying these processes.

I have only two relatively minor concerns.

The first involves the writing itself. This is a complicated paper, and unfortunately it does not help that there are numerous errors of English (singular vs plural, missing articles, past vs present tenses) or awkward phrasings, including a lot of run on sentences throughout the manuscript. Here is one such example:

Consistent with the previous study showing that expression of *Zglp1* mRNA reached peaks at E12.5 and E13.5 and declined afterward, and that expression of ZGLP1 started to appear at E12.5 and precedes that of STRA8 at THE protein level (Nagaoka et al. 2020), *Zglp1* and other downstream genes of BMP signaling (*Id1*, *Id2*, *Id3*, *Gata2*, *Msx1*, *Msx2*) were expressed in *Stra8*-null oogonia, albeit at variable levels (Fig. 2G, Fig. S5D) (RUN ON SENTENCE THAT IS VERY HARD TO DECIPHER). These data suggested that *Stra8*-null oogonia had at least received female specific sex determination (UNCLEAR MEANING ___ HOW DO OOGONIA "RECEIVE" FEMALE SPECIFIC SEX DETERMINATION) and responded to BMP signaling prior to meiosis. RA responsive genes such as *Rec8* (Koubova et al. 2006) WERE also expressed in *Stra8*-null oogonia, indicative of the response to RA (Fig. 2H, Fig. S5D). Altogether, STRA8 was required for THE timely exit from the pluripotent/PGC status to become competent for meiosis in female germ cells.

It would be extremely useful to the reviewer if the paper could be proofed to eliminate these problems.

The second involves the meiotic phenotypes of the STRA8 mutation. Clearly, the main purpose of the studies is to elucidate the roles of STRA8/RB/MEIOSIN in meiotic entry and the progression of cells through the early stages of meiosis. However, it seems like a real missed opportunity to skip any analyses of possible meiotic malformations. That is, it is clear from Fig 7 that the authors could examine meiosis-acting proteins (e.g., SYCP3, DMC1, MLH1, RAD51) by immunostaining to ask whether there are any obvious abnormalities in pairing, synapsis, or DSB production or repair in the mutants. These would be relatively easy to conduct, and would shed light on the possibility of specific meiotic malformations involving STRA8 mutants.

REVIEWER COMMENTS

Reviewer #1 (Remarks to the Author):

Key results

This team has previously revealed the importance of MEIOSIN, which interacts with the critical pre-meiotic transcription factor STRA8 and is necessary for meiotic onset and progression (Ishiguro et al 2020). Here they show that STRA8 also binds to RB1 and related protein p107, and that this binding is independent of the STRA8/MEIOSIN interaction and uses a different domain of STRA8. They use in vivo mutation of the evolutionarily-conserved RB1 binding site in STRA8 (referred to as LXCXE; X is replaced by Alanine) to reveal that STRA8/RB interaction ensures timely meiotic onset, in terms of both meiotic S phase entry and meiotic gene activation. They also show that mutation of the RB binding site leads to precocious depletion of the oocyte pool. The phenotype of the LXCXE mutant is, however, not as severe as the Stra8 KO (the latter fails to enter meiosis at all, whereas the former is delayed in meiotic progression). The model suggests that STRA8 sequesters RB from E2F at the G1/S transition, thereby ensuring pre-meiotic G1/S phase transition. It seems that nuclear localization of STRA8 depends on both RB1 (shown here) and MEIOSIN (shown previousl, Ishiguro et al 2020). The significance is potentially limited by the fact that meiotic onset and progression is only slowed in the absence of STRA8-RB1 complex formation.

Overall assesement

The quality of the data seems good. A great deal of work has been done and, technically, I have no reason to doubt that it is sound. The greatest problem with the manuscript is that it is very loosely written and hard to follow.

Major points to address

1. Do we know that STRA8 co-immunoprecipitates with RB and p107 in the fetal ovary?

We attempted to immunoprecipitate STRA8 from fetal ovaries. However, we found that it was technically difficult to show the STRA8-RB1/p107 interaction biochemically by IP-WB in fetal ovaries, because of the limited starting materials of fetal ovaries, unlike IP-WB using testes. The amount of immunoprecipitated STRA8 protein level was too low to detect co-immunoprecipitation of RB1/p107. Thus, we described this technical limitation in the main text (Line 121-123).

2. Why would some STRA8 bind MEIOSIN and some STRA8 bind RB? What would prevent all three being present in one complex?

Although we did not address mechanistic insight in this point, one possibility is that binding of RB or MEIOSIN may exclude interaction to another due to steric hindrance. We are currently collaborating with structural biologist to see mechanistic insight into the STRA8-MEIOSIN complexes, which awaits further structural analysis. We discussed this possibility in the main text (Line 117-120).

3. Why would MEIOSIN expression be weaker in the Stra8DLxCxE-3FHKI than in the Stra8 wt -3FH KI control (Figure 1D)? Couldn't this be the reason there is more cytoplasmic STRA8,

rather than a role of RB1 here? (lines 139 to 144)

In the Fig.1D, ovarian sections from *Stra8*^{wt}-3FH KI and *Stra8*^{ΔLxCxE}-3FH KI heterozygous were shown. Relatively weaker MEIOSIN expression level at this time point may also additively affect STRA8 nuclear localization in *Stra8*^{ΔLxCxE}-3FH KI heterozygous mice, we assume that this difference could be due to subtle differences in the developmental progression of the embryos between heterozygous *Stra8*^{wt}-3FH KI and *Stra8*^{ΔLxCxE}-3FH KI mice at E14.5, as we later showed that *Meiosin* expression is delayed in *Stra8*^{wt}-3FH KI homozygous ovaries (Fig. 4E and Fig. 5E).

Thus, we added the following sentences (lines 155 to 159); “It is worth noting that weaker MEIOSIN expression level at this time point may additively affect STRA8 nuclear localization in *Stra8*^{ΔLxCxE}-3FH KI heterozygous mice. We assume that this difference in MEIOSIN expression level between *Stra8*^{wt}-3FH KI and *Stra8*^{ΔLxCxE}-3FH KI mice at E14.5 could be due to subtle differences in developmental progression of the embryos (see discussion below, Fig. 4E and Fig. 5E).”

4. Why do you think *Zglp1* expression differs between the wildtype and the LxCxE mutant (Figure 2G)? This does not seem consistent with the idea that ZGLP1 is downstream of BMP signaling but not downstream of RA or STRA8.

Our scRNA-seq data suggests that *Zglp1* expression was overall downregulated during developmental progression and further decreased in pre-meiotic G1/S phase transition. Since germ cells were yet to reach pre-meiotic G1/S phase at E14.5 in the LxCxE mutant, residual level of *Zglp1* was apparently higher in LxCxE mutant compared to WT. Nevertheless, whereas *Zglp1* expression was independent of STRA8, its pattern seems to be somewhat transient compared to those of other downstream genes of BMP signaling such as *Msx1*.

Now, we added the following sentences (Line 206-211); “Although *Zglp1* expression was independent of STRA8, its pattern was transient compared to that of other downstream genes of BMP signaling, such as *Msx1*. Furthermore, whereas *Zglp1* expression declined in *Stra8* ΔLXCXE /ΔLXCXE compared to *Stra8*-null oogonia, the residual level of *Zglp1* was still higher in *Stra8* ΔLXCXE /ΔLXCXE than in WT. This suggests that the developmental progression of *Stra8* ΔLXCXE /ΔLXCXE germ cells advanced compared to *Stra8*-null oogonia, but was delayed compared to WT.”

5. Oct4 not included in Figure 2D and is not mentioned as a gene related to stem cell maintenance (line 170) - does it act differently to *Nanog*, *Sox2*, *Esrrb* and *Klf5*?

Although *Pou5f1* seems to be expressed in more differentiated cells compared to *Nanog*, *Sox2*, *Esrrb* and *Klf5*, the expression of *Pou5f1* declined over developmental progression and similar to those of other pluripotent marker genes.

Now we added expression profile of *Pou5f1* in the new Figure 2D.

6. The story line becomes a little muddled when the *Stra8* KO is included. I think it would be better to focus on the LxCxE mutant.

We agree to the reviewer's comment. However, we emphasize that the phenotype at transcriptome level is different between the *Stra8*^{ΔLxCxE} and the *Stra8* KO oocytes/oogonia. Thus, we would like to remain the story about *Stra8* KO (Figure 2) as it is.

7. Line 189 - you say that STRA8 is required for the germ cells to become 'competent for meiosis' but I do not see how this terminology is justified. In fact, the actions of BMP signaling might be more about 'competence' - in the absence of STRA8 the BMP targets are induced, but yet there is no meiosis as there is no STRA8.

We removed the sentence 'competent for meiosis', and instead replaced it with 'prior to meiosis' (Line 214).

8. Meiosin expression depends on the cell cycle status of G1/S transition in female germ cells (line 252) (what does this mean? that this is downstream of STRA8? this does not agree with earlier paper where it was claimed that both are induced by RA but in independent pathways?). Also MEIOSIN protein is lower in the LxCxE mutant. What does this mean?

MEIOSIN expression is independent of STRA8 and coincides with clusters 5 and 7 which represent G1/S transition. Now, to avoid confusion, we rephrased it as “*Meiosin* expression coincided with G1/S transition in female germ cells (Line 277 -280).

Because most of germ cells in the *Stra8*^{ΔLXCXE/ΔLXCXE} were yet to reach the G1/S phase transition (clusters 5 and 7), MEIOSIN expression was delayed and underrepresented in *Stra8*^{ΔLXCXE/ΔLXCXE} at E14.5. Now, we added the relevant sentences (Line 277 -280).

9. If STRA8 sequesters RB1 away from E2F to 'depress the expression of E2F-responsive genes' then why is the expression of this gene set lower in the LXCXE mutant? (Figure 3F)?

We are afraid that the reviewer seems to misunderstand and interpret that sequestering RB1 away from E2F by STRA8 to '**depress**' the expression of E2F-responsive genes', Correctly, this should be read that sequestering RB1 away from E2F by STRA8 to '**derepress**' the expression of E2F-responsive genes'.

Minor points to address

1. The first sentence of the abstract is a little obtuse - 'by sexually different mechanisms in mice'. This is also stated line 45. Suggests that the process of meiosis is different in different sexes, rather than the timing of meiosis onset.

Now we rephrased “by sexually different mechanisms in mice” to “whereby the timing of meiosis onset is different in male and female” in the abstract and the main text (Line 48).

2. LXCXE is often written as LxCxE. It would be good if one version could be used throughout, including in the figures. Why not LACAEE?

We appreciate the reviewer to point out that our way of describing the mutant allele may cause a confusion. Alanine substitutions were introduced at the Leu, Cys and Glu residues in the LXCXE motif of STRA8, so that “LKCLE” residues in WT were changed to “AKALA” in the mutant. Because nomenclature of LXCXE motif is well recognized as RB pocket domain binding motif in the cell cycle research field, “*Stra8*^{ΔLXCXE}” rather than “*Stra8*^{ΔKALA}” would easy to remind readers of that RB pocket domain binding motif was disrupted in the mutant. “Now we revised “*Stra8*^{ΔLxCxE}” to *Stra8*^{ΔLXCXE}” throughout all text and figures.

3. In general, the referencing is sloppy. References presented in Introduction line 40 seem quite arbitrary (eg Oulad-Abdelghani 1996 and Menke 2003 did not link STRA8 and

meiotic entry).

Later at line 53/54, Saitou and Yamaji 2012 and Yokoboyashi et al 2013 reference 'downregulation of pluripotency genes' at the time of meiotic initiation (but the first of these does not mention this and the second only in passing). Line 57 - not sure that Cordeiro et al 2015 is appropriate here (this ref about follicle activation not meiotic onset) - there are a number of other references where the anterior-posterior wave were revealed. There are many other examples - it is important to be precise in the literature. Reference the primary publications where possible and reviews where necessary, preferably noting that it is a review.

We corrected this.

4. 'ALDH1a families' is not clear

We correct this to ALDH1A proteins (ALDH1A1, 2, and 3) to describe the reference precisely.

5. Line 68 - 'how initiation of meiosis is coordinated with cell cycle so that it coincides with pre-meiotic S phase' - wouldn't it be inducing pre-meiotic S phase rather than just coinciding?

Now, we rephrased it as "how initiation of meiosis coincides with cell cycle".

6. Line 75 - 'Here we present sexually different mechanisms of meiotic initiation in mice' doesn't seem to make sense, nor to overview the findings of the study.

Now, we rephrased it as "female-specific mechanisms of meiotic initiation" (Line71).

7. Figure 1B has both Stra8DRb-3FHKI as well as Stra8DLxCxE-3FHKI shown (I think these are the same thing, and the second more consistent with the manuscript).

Now, we corrected this to *Stra8*^{ALXCXE}.

8. Throughout, Rb should be referred to by its correct name, Rb1

Now, we corrected this throughout the text and Figures. In the cases that both RB1 and p107 as Retinoblastoma family members are stated in the context, we referred RB1 and p107 together to as RB.

9. Line 116 - The expression of Stra8 in the testis was first shown by Oulad-Abdelghani 1996.

We add this reference.

10. Line 133 - I do not think you can say that 'FLAG-HA-tagged STRA8 was expressed at similar timing from both Stra8 wt -3FH KI and Stra8 ΔLxCxE-3FH KI alleles' when you only looked at a solitary timepoint, E14.5.

Now, we removed "at similar timing" and simply rephrased it as "the FLAG-HA-tagged STRA8 was expressed from both *Stra8*^{wt} -3FH KI and *Stra8*^{ΔLXCXE} -3FH KI alleles" (Line143).

11. Line 167 - I am not sure that you can say 'distinct clusters'. This would be true of the wildtype vs KO, but the wildtype vs LxCxE homs seem to reveal different clustering but not 'distinct clusters'.

Now, we rephrased it as "different clusters" (Line182).

12. Figure 2B - says verocity not velocity

Now, we corrected this.

13. I am not sure why it would be 'remarkable' that pluripotency-associated gene expression would remain high in the *Stra8* KO cells? I think this is similar to observations by many other studies (eg see the reciprocal expression patterns in Bullejos and Koopman 2004 *Mol Repro and Dev*).

Now we rephrased it as “Consistent with a previous study” and added the reference (Bullejos and Koopman 2004).

14. Wu et al 2016 does not belong here - this was regarding SMAD4 deletion not BMP action.

A reference Wu et al 2016 was removed.

15. Line 195 - says 11 clusters, but only ten are shown in Figure S5.

Cluster numbering starts from 0 to 10. Thus, there are 11 clusters.

16. Figure 3G - why is quantitation not shown for this particular gene as it is for the others?

Now quantitation is shown for Figure 3G.

17. Line 307 - the word 'between' here is not appropriate (suggests communication?). 'Gene expression changes as primordial follicles transition to primary follicles'?

We revised this to “..dynamic gene expression changes as primordial follicle transition to primary follicles” as suggested by the reviewer.

18. Line 26, 309, 328, 364, 375 - not sure what you mean by 'unscheduled' - sounds like ectopic or early but I think you mean abnormal or delayed.

All the 'unscheduled' was changed to 'delayed'.

19. Line 310 - 'Oocytes accompanied' should possibly be 'oocytes displayed'

Now, we revised this.

20. Line 328 - why would we consider this an 'indirect consequence'? Surely if delayed at E14.5 this might lead to delay at E18.5?

Now, we simply rephrased it as “as a consequence of delayed entry into meiosis and subsequent delayed progression of meiotic prophase” (Line359).

21. Line 341 - this implies that the *Stra8* KO is similar to the *Stra8* LxCxE mutant in terms of fertility phenotype - but I think the follicles are already absent at birth for the former?

In the present study, we did not perform comparable analysis of *Stra8* KO ovaries together with *Stra8*^{ΔLXCXE} mutant to see whether the follicles are present or absent at birth. Although we do not know the follicles are already absent at birth in *Stra8* KO ovaries, oocytes loss would be accelerated in *Stra8* KO ovaries (Dokshin et al. 2013).

22. Line 343 - 'starved'? lost?

We corrected this to “lost”.

23. Line 396 - additional and original reference is Bowles et al, 2006

We added this reference.

24. Line 397 - I do not understand what you mean by 'this could presumably be due to actively driving cell cycle in the passage from spermatogonia to spermatocyte, which is in contrast to the oogonia'.

Now, we rephrased it as “This could presumably be because de-repression of E2F responsive genes is mediated by phosphorylation of RB1 at pre-meiotic S phase in male.”

25. Line 419 - you seem to have left out the step where p16/p19 antagonise CDKs

We added the sentence “Since p16^{INK4a} and p19^{ARF} directly or indirectly lead to antagonize CDKs,”

26. Line 421 - 'the overall expression levels of E2F-target genes were relatively lower in pre-meiotic G1/S (Cluster 5) compared to mitotic G1/S (Cluster 6) - Fig 3A, E.' I am very confused - Figure 3 doesn't show clusters at all. And overall expression of E2F target genes is lower in the LXCXE mutant in Figure 3F.

To avoid a confusion, now we removed the sentence “Consistent with this notion, the overall expression levels of E2F-target genes were relatively lower in pre-meiotic G1/S (Cluster 5) compared to mitotic G1/S (Cluster 6) (Fig. 3A, E)”.

27. Expression level of the *Cdkn2a* was reciprocally higher in pre-meiotic G1/S than in mitotic G1/S (Fig. 3F) - ??? this figure compares Stra8 Wt with the LXCXE mutant??

As mentioned above (26), to avoid a confusion now we removed this sentence “while the expression level of the *Cdkn2a* was reciprocally higher in pre-meiotic G1/S than in mitotic G1/S (Fig. 3F)”.

28. There is no Figure 4F shown.

Fig 4F is presented.

Reviewer #2 (Remarks to the Author)

This is a detailed analysis of factors involved in initiation and progression of meiosis in female mice. Combining transcriptomic analyses with different STRA8 mutations (most importantly, one affecting STRA8-RB binding), the authors provide an exhaustive analysis of the regulation of meiotic initiation in mammalian females and its progression through the fetal stages of meiosis. This adds important new insights to the way in which oocytes are formed, and the authors are to be congratulated on characterizing the complicated set of interactions underlying these processes.

I have only two relatively minor concerns.

The first involves the writing itself. This is a complicated paper, and unfortunately it does not help that there are numerous errors of English (singular vs plural, missing articles, past vs present tenses) or awkward phrasings, including a lot of run on sentences throughout the manuscript. Here is one such example:

Consistent with the previous study showing that expression of *Zglp1* mRNA reached peaks at E12.5 and E13.5 and declined afterward, and that expression of ZGLP1 started to appear at E12.5 and precedes that of STRA8 at THE protein level (Nagaoka et al. 2020), *Zglp1* and other downstream genes of BMP signaling (*Id1*, *Id2*, *Id3*, *Gata2*, *Msx1*, *Msx2*) were expressed in *Stra8*-null oogonia, albeit at variable levels (Fig. 2G, Fig. S5D) (RUN ON SENTENCE THAT IS VERY HARD TO DECIPHER). These data suggested that *Stra8*-null oogonia had at least received female specific sex determination (UNCLEAR MEANING __ HOW DO OOGONIA "RECEIVE" FEMALE SPECIFIC SEX DETERMINATION) and responded to BMP signaling prior to meiosis. RA responsive genes such as *Rec8* (Koubova et al. 2006) WERE also expressed in *Stra8*-null oogonia, indicative of the response to RA (Fig. 2H, Fig. S5D). Altogether, STRA8 was required for THE timely exit from the pluripotent/PGC status to become competent for meiosis in female germ cells.

It would be extremely useful to the reviewer if the paper could be proofed to eliminate these problems.

Now we had our revised manuscript proofed to eliminate English problems.

The second involves the meiotic phenotypes of the STRA8 mutation. Clearly, the main purpose of the studies is to elucidate the roles of STRA8/RB/MEIOSIN in meiotic entry and the progression of cells through the early stages of meiosis. However, it seems like a real missed opportunity to skip any analyses of possible meiotic malformations. That is, it is clear from Fig 7 that the authors could examine meiosis-acting proteins (e.g., SYCP3, DMC1, MLH1, RAD51) by immunostaining to ask whether there are any obvious abnormalities in pairing, synapsis, or DSB production or repair in the mutants. These would be relatively easy to conduct, and would shed light on the possibility of specific meiotic malformations involving STRA8 mutants.

We performed cytological analyses to see possible meiotic phenotypes in the *Stra8*^{ΔLXCXE/ΔLXCXE} oocytes. The results indicated that ;

(1) Transition of the number of DMC1 foci across the stages of meiotic prophase was comparable between the control and *Stra8*^{ΔLXCXE/ΔLXCXE} oocytes. This suggests that at least generation of DSBs, and meiotic recombination were proficient in *Stra8*^{ΔLXCXE/ΔLXCXE} oocytes. Now we added the new data in the Fig. 5i. Line 332-337.

(2) MLH1 foci were observed in *Stra8*^{ΔLXCXE/ΔLXCXE} pachytene oocytes. This suggests that crossover recombination at least in part seems to be proficient in *Stra8*^{ΔLXCXE/ΔLXCXE} oocytes, though slight decrease in the number of MLH1 foci may be due to delayed progression of meiotic prophase. Now we added the new data in the Fig. 5J. Line 336-342.

(3) Fig 5h indicates *Stra8*^{ΔLXCXE/ΔLXCXE} oocytes showed homolog synapsis and progressed through the meiotic prophase, as assessed by immunostaining of SYCP1.

These lines of data suggest that despite a delay in meiotic initiation, once the *Stra8*^{ΔLXCXE/ΔLXCXE} oocytes enter into meiotic prophase, homolog synapsis, meiotic HR recombination and crossover are

apparently proficient in *Stra8*^{ΔLXCXE/ΔLXCXE} oocytes. This idea is consistent with the fact that 20 pairs of bivalents with chiasmata were observed in *Stra8*^{ΔLXCXE/ΔLXCXE} Meta I oocytes (Fig. 7c). The presence of chiasmata in *Stra8*^{ΔLXCXE/ΔLXCXE} oocytes indicated that they had progressed through the meiotic prophase and had experienced meiotic recombination processes.

Since scRNA-seq suggested that upregulation of those genes of p53-dependent damage response in *Stra8*^{ΔLXCXE/ΔLXCXE} oocytes at E18.5 (Supplementary Fig. 7e and f), we do not exclude a possibility that p53 dependent response might in part derive from a hidden meiotic defect in the processes of meiotic recombination and homolog synapsis. Now, we added discussion on this possibility (line 450-464).

Reviewer #1 (Remarks to the Author):

Only major point is the way the story is presented - that this STRA8-RB and/or STRA8-MEIOSIN interaction mechanism somehow coordinates meiotic initiation and S phase progression. Overall, if the thesis is correct - 'mechanisms executed by the STRA8-RB and STRA8-MEIOSIN subcomplexes coordinate the synchronization of S phase progression and meiotic gene activation for timely meiotic entry in female germ cells' - then wouldn't we expect this specific mutation of STRA8 sequence to affect only the S phase progression and not the meiotic gene activation? But the whole meiotic process seems to be delayed. In particular, expression of key genes Meiosin and Zglp1, as well as p107 expression, is delayed - and we would probably not expect these to be E2F targets?

The first sentence of the Abstract doesn't seem to make sense? 'Meiosis coordinates with the developmental program of germ cells'.

The confusing 'Sexually different mechanisms of meiotic initiation' remains - as the Running Title. I am not sure what this actually means?

A good running title could be 'STRA8 directly interacts with RB at meiotic onset'.

Line 62 - 'by' should be replaced by a comma. There are two Bowles et al 2006 references here?

Line 67. The section starting 'In addition to RA signaling' does not acknowledge that the 'female germ cells' are actually in vitro derived PGCLCs and this has not been shown 'in females'.

Line 158 - this all sounds a bit weak? (in the red). Since MEIOSIN not detected until E15.5 (unlike E14.5 in the WT) the difference is subcellular localisation could be entirely to do with a lack of MEIOSIN I think (it is not possible to conclude that it has anything to do with the STRA8 mutation?).

Line 176/177 - were generated/were used - redundant

Line 202 - reference Miyauchi et al 2017 appears twice.

Line 209 - rewrite like this?

Zglp1 expression is independent of STRA8 and its pattern of expression is transient, compared to other downstream genes of BMP signaling, such as Msx1 (Nagaoka et al, 2020). In our studies, we find Zglp1 expression remains relatively high in the Stra8-null oogonia, compared with WT, and that Zglp1 expression is intermediate in the Stra8 Δ LXCXE/ Δ LXCXE mutant oogonia. This suggests that the developmental progression of Stra8 Δ LXCXE/ Δ LXCXE oogonia is intermediate between WT and Stra8-null oogonia.

Line 247 - Expression of p107 was 'less' seems like a bit of an understatement (it is essentially gone, compared to WT?)

Line 274 - you say cluster 5 or 6 but I think you mean 5 or 7?

Line 280 - doesn't this indicate that Meiosin expression depends on the intact STRA8 protein and its interaction with RB?

Line 322 - if MEIOSIN protein is not found until E15.5 in the Stra8 mutant, though usually is seen at E14.5, then it does not seem appropriate to conclude that 'nuclear localization of STRA8 depends on both RB and MEIOSIN in female germ cells'. How do we know it is not simply that STRA8 depends on MEIOSIN for nuclear localisation?

Line 410 - I don't think the word 'indirect' is warranted here.

Line 440 - should be 'in' not 'into'

Line 448 - 'These lines of evidence account for the strict requirement of STRA8-RB interaction for meiosis entry in mouse females' - this seems a stretch - a 'strict' requirement suggests that something just does not happen, whereas here we just see a delay?

Line 512 - 'As Stra8 Stra8 Δ LXCXE/ Δ LXCXE oogonia responded to RA signal at E14.5, Meiosin expression in female germ cells might be presumably regulated indirectly or independently of RA rather than simply downstream of RA' - This is very confusing.

Maybe 'As Stra8 Stra8 Δ LXCXE/ Δ LXCXE oogonia are clearly capable of responding to the endogenous RA signal at E14.5 (Fig. 2h; normal expression of Rec8), it seems that Meiosin expression in female germ cells is not directly regulated by RA, as previously proposed (Ishiguro et al, 2020).

Rebuttal

Maybe also mention, in the Discussion, the caveat that STRA8/RB interaction has not been demonstrated yet in the fetal ovary.

REVIEWERS' COMMENTS

Reviewer #1 (Remarks to the Author):

Only major point is the way the story is presented - that this STRA8-RB and/or STRA8-MEIOSIN interaction mechanism somehow coordinates meiotic initiation and S phase progression. Overall, if the thesis is correct - 'mechanisms executed by the STRA8-RB and STRA8-MEIOSIN subcomplexes coordinate the synchronization of S phase progression and meiotic gene activation for timely meiotic entry in female germ cells' - then wouldn't we expect this specific mutation of STRA8 sequence to affect only the S phase progression and not the meiotic gene activation? But the whole meiotic process seems to be delayed. In particular, expression of key genes Meiosin and Zglp1, as well as p107 expression, is delayed - and we would probably not expect these to be E2F targets?

As suggested by the reviewer, we replaced the relevant sentences as following

Accordingly, the title was changed to “STRA8–RB interaction is required for timely entry of meiosis in mouse female germ cells”.

Line 32: “STRA8–RB interaction is required for S phase entry and meiotic gene activation in female germ cells”.

Line 292: “meiotic gene activation coincides with the G1/S transition of the cell cycle”

Line 341 : “the STRA8–RB interaction is required for ~~coordinates the synchronization of~~ pre-meiotic S phase entry and meiotic gene activation”

Line 411 subheading : STRA8–RB interaction is required for G1/S transition and meiotic initiation in female germ cells

The first sentence of the Abstract doesn't seem to make sense? 'Meiosis coordinates with the developmental program of germ cells'.

We replaced the sentence as following “Meiosis is differently regulated in males and females.”

The confusing 'Sexually different mechanisms of meiotic initiation' remains - as the Running Title. I am not sure what this actually means?

A good running title could be 'STRA8 directly interacts with RB at meiotic onset'

The Running Title was removed since this journal does not require the Running Title.

Line 62 - 'by' should be replaced by a comma. There are two Bowles et al 2006 references here?

We corrected this.

Line 67. The section starting 'In addition to RA signaling' does not acknowledge that the 'female germ cells' are actually in vitro derived PGCLCs and this has not been shown 'in females'.

We rephrased this sentence as follows.

In vitro study suggested that meiotic initiation is also regulated by several regulatory signals in female PGC-like cell (PGCLC).

Line 158 - this all sounds a bit weak? (in the red). Since MEIOSIN not detected until E15.5 (unlike E14.5 in the WT) the difference is subcellular localisation could be entirely to do with a lack of

MEIOSIN I think (it is not possible to conclude that it has anything to do with the STRA8 mutation?).

We removed this sentence.

Line 176/177 - were generated/were used - redundant

We corrected this.

Line 202 - reference Miyauchi et al 2017 appears twice.

We replaced this reference with (Nagaoka et al. 2020).

Line 209 - rewrite like this?

Zglp1 expression is independent of STRA8 and its pattern of expression is transient, compared to other downstream genes of BMP signaling, such as Msx1 (Nagaoka et al, 2020). In our studies, we find Zglp1 expression remains relatively high in the Stra8-null oogonia, compared with WT, and that Zglp1 expression is intermediate in the Stra8 Δ LXCXE/ Δ LXCXE mutant oogonia. This suggests that the developmental progression of Stra8 Δ LXCXE/ Δ LXCXE oogonia is intermediate between WT and Stra8-null oogonia.

We revised the sentences as the reviewer suggested.

Line 247 - Expression of p107 was 'less' seems like a bit of an understatement (it is essentially gone, compared to WT?)

We thought that 'gone' is overstatement only based on the scRNA-seq data since weaker levels of gene expression cannot be detected by scRNA-seq.

Line 274 - you say cluster 5 or 6 but I think you mean 5 or 7?

Velocity analysis indicates two directions towards cluster 5 (pre-meiotic S) and cluster 6 (mitotic S). Thus, we mean cluster 5 or 6.

Line 280 - doesn't this indicate that Meiosin expression depends on the intact STRA8 protein and its interaction with RB?

This not necessarily mean Meiosin expression depends on the intact STRA8 protein and its interaction with RB, since it is known that Meiosin is expressed independently of STRA8 in male (Ishiguro et al, Dev 2020).

Line 322 - if MEIOSIN protein is not found until E15.5 in the Stra8 mutant, though usually is seen at E14.5, then it does not seem appropriate to conclude that 'nuclear localization of STRA8 depends on both RB and MEIOSIN in female germ cells'. How do we know it is not simply that STRA8 depends on MEIOSIN for nuclear localisation?

We used E15.5 ovary for this analysis. To avoid confusing the reader, we added the information of sampling stage in Figure legend.

Even in the absence of MEIOSIN, STRA8 tend to be localized in cytoplasm in Stra8 Δ LXCXE mutant compared with control. Therefore, we concluded that STRA8 localization depends on both RB and MEIOSIN.

Line 410 - I don't think the word 'indirect' is warranted here.

We removed “indirect”.

Line 440 - should be 'in' not 'into'

We corrected.

Line 448 - 'These lines of evidence account for the strict requirement of STRA8-RB interaction for meiosis entry in mouse females' - this seems a stretch - a 'strict' requirement suggests that something just does not happen, whereas here we just see a delay?

We removed “strict”.

Line 512 - 'As *Stra8* *Stra8* Δ LXCXE/ Δ LXCXE oogonia responded to RA signal at E14.5, Meiosin expression in female germ cells might be presumably regulated indirectly or independently of RA rather than simply downstream of RA' - This is very confusing.

Maybe 'As *Stra8* *Stra8* Δ LXCXE/ Δ LXCXE oogonia are clearly capable of responding to the endogenous RA signal at E14.5 (Fig. 2h; normal expression of Rec8), it seems that Meiosin expression in female germ cells is not directly regulated by RA, as previously proposed (Ishiguro et al, 2020).

We rephrased it as the following “As *Stra8* ^{Δ LXCXE/ Δ LXCXE} oogonia are capable of responding to the endogenous RA signal at E14.5 (Fig. 2h), it seems that *Meiosin* expression in female germ cells is not directly regulated by RA.”, as suggested by the reviewer.

Rebuttal

Maybe also mention, in the Discussion, the caveat that STRA8/RB interaction has not been demonstrated yet in the fetal ovary.

We added following sentence.

Due to the limited availability of materials, it was technically difficult to show the interaction between STRA8 and RB by IP-MS experiment using extracts from the embryonic ovary although we tried to do it. (Line417- 419)